# The lateralization of left hippocampal CA3 during the retrieval of spatial working memory

Da Song[1,6], Deheng Wang [2,6], Qinghu Yang [1,6], Tianyi Yan[1,6], Zhe Wang[3], Yan Yan[1], Juan Zhao[1], Zhen Xie[1], Yuchen Liu[2], Zunji Ke [2], Talal Jamil Qazi [1], Yanhui Li[1], Yili Wu[4], Qing Shi [5], Yiran Lang [5], Heao Zhang[1], Tao Huang[1], Chunjian Wang[1], Zhenzhen Quan [1✉] & Hong Qing [1✉]

The hippocampal CA3 contributes to spatial working memory (SWM), but which stage of SWM the CA3 neurons act on and whether the lateralization of CA3 function occurs in SWM is also unknown. Here, we reveal increased neural activity in both sample and choice phases of SWM. Left CA3 (LCA3) neurons show higher sensitivity in the choice phase during the correct versus error trials compared with right CA3 (RCA3) neurons. LCA3 initiates firing prior to RCA3 in the choice phase. Optogenetic suppression of pyramidal neurons in LCA3 disrupts SWM only in the choice phase. Furthermore, we discover that parvalbumin (PV) neurons, rather than cholinergic neurons in the medial septum (DB were cholinergic neurons), can project directly to unilateral CA3. Selective suppression of PV neurons in the MS projecting to LCA3 impairs SWM. The findings suggest that MS$^{PV}$-LCA3 projection plays a crucial role in manipulating the lateralization of LCA3 in the retrieval of SWM.

[1] Key Laboratory of Molecular Medicine and Biotherapy, School of Life Science, Beijing Institute of Technology, Beijing 100081, China. [2] School of Basic Medicine, Shanghai University of Traditional Chinese Medicine, Shanghai 201203, China. [3] Advanced Innovation Center for Human Brain Protection, Capital Medical University; The National Clinical Research Center for Geriatric Disease, Xuanwu Hospital, Capital Medical University, Beijing, China. [4] Shandong Collaborative Innovation Center for Diagnosis & Treatment and Behavioral Interventions of mental disorders & Shandong Key Laboratory of Behavioral Medicine, Institute of Mental Health, Jining Medical University, Jining, Shandong 272067, China. [5] Beijing Advanced Innovation Center for Intelligent Robots and Systems, Beijing Institute of Technology, Beijing 100081, China. [6] These authors contributed equally: Da Song, Deheng Wang, Qinghu Yang, Tianyi Yan. ✉email: qzzbit2015@bit.edu.cn; hqing@bit.edu.cn

One of the most predominant and fundamental role of hippocampus is illustrated as the formation of a cognitive map that manipulates different sorts of learning and memory both in humans and in nonhuman species[1]. In human, the hippocampus is considered anatomically symmetrical, but functionally lateralized (especially in task-related activities). It is reported that the right hippocampus controls spatial information processing and the left takes charge of verbal semantic representations[2,3]. A functional MRI study suggests that two hemisphere hippocampi perform a complementary principle on navigation with places information processing on the right and temporal sequences on the left[4]. In a study of neurosurgical patients, hippocampus is shown to have lateralized oscillatory patterns in responsive to memory encoding and navigation[5]. Thus, hippocampal lateralization in spatial cognition is a common phenomenon in higher-order brain function.

Actually, this left-right asymmetry in hippocampal circuitry also occurs in rodents species[6–8]. It has demonstrated that asymmetrical deficits of hippocampal circuitry caused by asymmetrical distribution of NMDA receptor GluRε2 subunits could impair working memory in several types of mouse models[9,10]. Moreover, the hippocampal CA3-CA1 pyramidal neuron synapses are revealed asymmetrical that inputs from RCA3 to CA1 possess larger and more perforated postsynaptic densities and GluR1 expression than that from LCA3, and the lateralization of right hippocampus is further functionally implicated to be responsible for improving the accuracy of spatial memory in animal behavioral tests[11,12]. Regarding the asymmetry of CA3-CA1 hippocampal circuitry, optogenetic application further determines inputs from LCA3 to CA1 lead to more long-term potentiation (LTP) than that from RCA3 due to differential GluN2B subunit-containing NMDARs at synapses. Besides, a high-frequency induced LTP is strengthened only when presynaptic input is originated from LCA3, which is crucial to spatial long-term memory[7,8]. However, whether the hippocampal CA3 neurons from left to right hemisphere have asymmetrical firing patterns in the working memory process and whether the lateralization is regulated by upstream projecting region to CA3 is still unknown.

Hippocampal CA3 is extensively involved in modulation of encoding cues in SWM, which is featured by storing and processing messages for goal-directed actions in cognition process[13,14]. Lesions of dorsal CA3, not dorsal CA1, impair the performance of SWM tasks[15,16]. It is recently observed that the dentate gyrus (DG) induced CA3 sharp-wave ripple supports reward-induced SWM[17]. However, which exact phase the CA3 neurons act on during the SWM process remains unclear.

To address these issues, we apply fiber photometry in the T-maze delay-no-match-to-place (tDNMTP) task and record the neural activity of subpopulations in different phases of SWM in mice. We reveal that CA3 neurons show increased neural activity in both the sample and the choice phases. More importantly, LCA3 and RCA3 neurons exhibit asymmetrical amplitudes of the increased neural activity in the choice phase during the correct versus the error trials. Single-unit recording demonstrates that LCA3 neurons tend to fire earlier than RCA3 neurons. Optogenetic stimulation of unilateral CA3 suggests LCA3 dominates spatial working memory. Different SWM-related behavioral tasks display the same consequences via optogenetic stimulation. Through combining rabies retrograde monosynaptic tracing and AAVretro retrograde system, we observe that PV and cholinergic neurons in DB were cholinergic neurons/DB have different projection patterns to CA3. MS[PV] neurons tend to project unilaterally to CA3. We also demonstrate that selective suppression of MS[PV]-LCA3 projecting neurons impairs the performance of SWM in the choice phase.

Together, our data support the lateralization of LCA3 neurons in the retrieval of SWM, and as such the influence of LCA3 in T-maze task is transferable to other diagrams of SWM tasks. MS[PV]-LCA3 projection is highlighted in CA3 lateralization in SWM.

## Results

**Hippocampal CA3 neural activity is lateralized during SWM.** To assess the neural activity of hippocampal CA3 neurons, we expressed the genetically encoded calcium indicator GCaMP6s in CA3 neurons (Fig. 1a). These neurons reflect action potential firing via fluorescence intensity during fiber photometry recording. The tDNMTP task was employed to test the animals' SWM[18,19]. To measure the behavioral performance, each trial has three phases by sequence: the sample phase, the delay phase, and the choice phase. During the sample phase, one of the two goal arms was blocked and the mouse was allowed to enter the open arm to get the food rewards. When the mouse initiated the sample run with the main arm open, the time point was marked as "sample begin". The delay phase was defined as 10 s by which the mouse was forced to stay after it returned to the main arm. During the choice phase, with the door removed, the mouse had to choose the arm opposite to the sample phase in order to receive the reward. The time was marked as "choice begin" when the mouse started the choice run.

For well-trained mice, the fluorescent intensities of the GCaMP6 signal were increased in both the sample and the choice runs, especially when the mice initiated to run after the main arm door was reopened. The signal remained unchanged during the 10 s delay phase (Fig. 1b and Supplementary Fig. 1h), which indicated CA3 neurons are activated during the memory encoding and retrieval phases rather than the delay phase.

As mice were trained from day 1 to day 8, mice achieved stable performance after 4 days of training, with an average success rate of 80%. Interestingly, the CA3 neural activity during the sample and delay phases changed gradually among task days (Fig. 1f). During the first day of the task, only minor increases of $Ca^{2+}$ signal were recorded during the sample and the choice phases. In response to the sample and the choice runs, peaks of increased neural activity gradually increased along with the progress of the task. They reached a plateau after five days of training, which produced a more abundant and stable $Ca^{2+}$ signal (Fig. 1e and Supplementary Fig. 1f). The amplitudes of the $Ca^{2+}$ activity in CA3 neurons correlated tightly with accumulated experience as the days proceeded (Fig. 1f). In addition, the $Ca^{2+}$ activity was proportional to the success rates (Fig. 1g), suggesting a correlation between a mouse's CA3 neural activity and working memory performance.

We further analyzed the differences between unilateral CA3 activity in the correct and error trials during the SWM task. In the sample and delay phases, mice in the LCA3- and RCA3-recorded groups showed similar $Ca^{2+}$ amplitude changes between the correct and error trials. Although both LCA3 and RCA3 neurons showed increased amplitude of $Ca^{2+}$ signal in the choice phase regardless of correct and error trials, a significant difference was found in LCA3 between the correct versus the error trials (Fig. 1c, d). LCA3 neurons showed more increased amplitude of $Ca^{2+}$ signal in the correct trials than that in the error trials. This suggested the lateralization of CA3 neurons occurs in the process of SWM, especially LCA3 neurons perform higher neuronal activity in making correct choices.

To exclude the false positive results caused by optical fiber movement, we recorded the eGFP expression in mice, which reflects the activity of CA3 neurons during free movement, and found it unchanged in the T-maze task (Supplementary Fig. 1a, b and e). The CA3 region is also responsible for encoding spatial

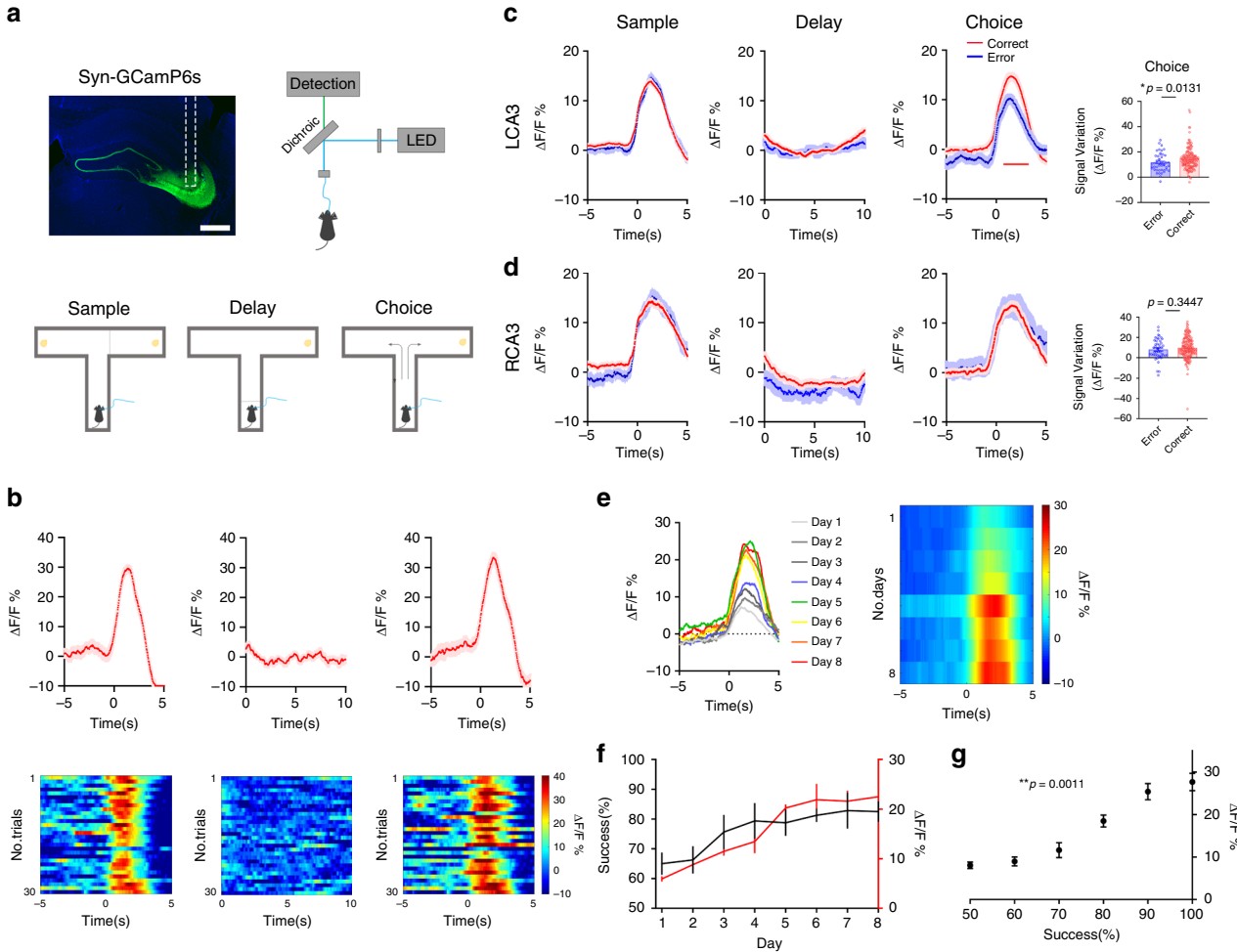

**Fig. 1 LCA3 neurons show higher amplitude of increased activity in T-maze task. a** Schematic of fiber photometry recording in T-maze task. Scale bar: 200 μm. **b** The mean and SEM of an example of corresponding $Ca^{2+}$ signal in CA3 region of a well-trained mouse at the sample, delay and choice phases. CA3 neurons were activated during the sample and choice phases, while remained unchanged during the delay phase. **c** The mean and SEM of $Ca^{2+}$ signal between the correct and error trials in LCA3 neurons at the sample, delay and choice phases from eight mice. The amplitude of increased signals in the correct trials was significantly higher than that in the error trials as mice initiated to run in the choice phase. Red line indicated bins between the correct and error trials with two-tailed unpaired t-test significance ($P < 0.05$). Right: quantification of the average amplitude of increased signal against baseline between the correct and error trials in LCA3 at the choice phase (error: $n = 49$ trials; correct: $n = 160$ trials; unpaired two-tailed Mann–Whitney-U-test, $*p = 0.0131$). **d** The mean and SEM of $Ca^{2+}$ signal between the correct and error trials in RCA3 neurons at the sample, delay and choice phases from six mice. The amplitude of increased signal had no differences between the correct and error trials in RCA3 in the choice phase. Right: quantification of the average amplitude of increased signal against baseline between the correct and error trials in LCA3 at the choice phase (error: $n = 47$ trials; correct: $n = 169$ trials; unpaired two-tailed Mann–Whitney-U-test, $p = 0.3447$). **e** The mean and SEM of $Ca^{2+}$ signal among 8 days and their corresponding heatmap. The amplitude of $Ca^{2+}$ signal was increased within processed days and reached a plateau after 5 days of training. **f** Plot of the amplitude of $Ca^{2+}$ signal and their corresponding success rates. **g** The correlation between SWM task performance and the amplitudes of $Ca^{2+}$ signal (two-tailed Pearson (R) correlation, $**p = 0.0011$, $r = 0.9726$). Data are presented as mean ± SEM (**c** and **d** right panels, **f**, **g**). Source data are provided as a Source Data file.

information. To exclude the possibility that the increasing activity of CA3 neurons was due to specific spatial location (like the T-junction crossing) rather than the task that the mice were performing, we allowed mice running in the T-maze for free movement and exploration. At each time point that a mouse run across the unclosed door in the main arm and intended to go through two goal arms, we measured the GCamP6 response. The results did not show significant changes of GCamP6 signal (Supplementary Fig. 1c), suggesting that the high increase of CA3 neural activity was irrelevant to the location that the mice pass through. Furthermore, we recoded $Ca^{2+}$ signal from free-running mice with a barrier (for 10 s) at the start point of the main arm and found that under non working-memory related task, the mice did not show any increased fluorescent signal (Supplementary Fig. 1d). Also, the signal to noise ratio from day 2 to day 8 remained stable,

which could exclude the influence of photodamage or bleaching (Supplementary Fig. 1g). These data implicated that the fluorescent signal is working-memory task dependent.

**The bilateral CA3 neural dynamics are defined in tDNMTP task.** To investigate the dynamics of a single neuron in bilateral CA3 during SWM task, we performed single-unit recording of mice performing the tDNMTP task to examine the different firing patterns between LCA3 and RCA3 neurons. We managed to record 227 and 197 well-isolated neurons from LCA3 and RCA3 respectively in 6 mice in the choice phase. We recorded the responses of neurons in LCA3 and RCA3 and classified neurons based on the direction, time point, and the magnitude of responses to the choice phase initiation in T-maze task (Fig. 2).

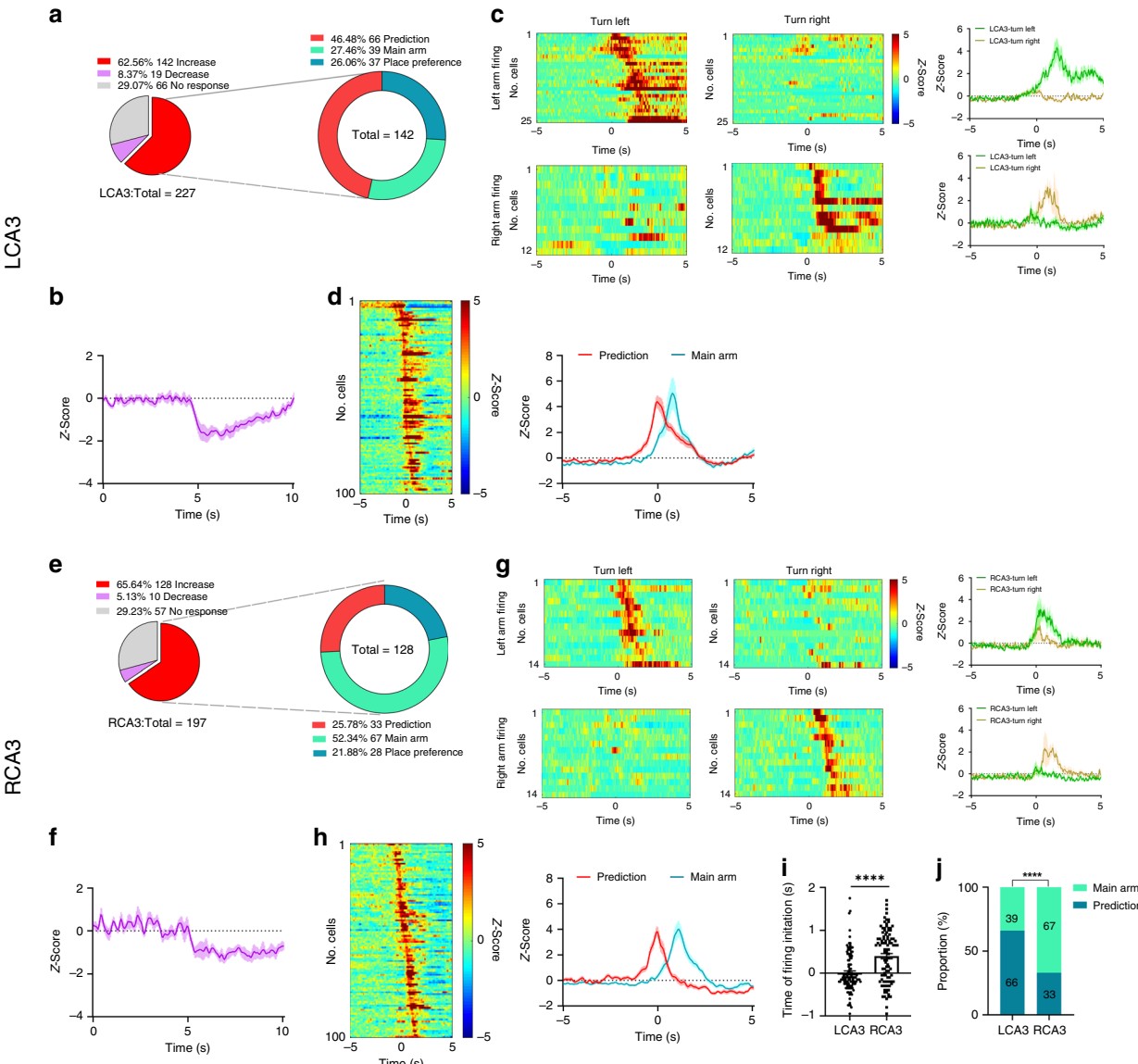

**Fig. 2 LCA3 neurons initiate firing earlier than RCA3 in the choice phase. a**, **e** The choice phase responsive neuron subpopulations in LCA3 or RCA3. Task-modulated neurons in LCA3 showed a greater proportion of responses when the choice phase initiated. On the right, task-activated neurons were differed into three types: prediction firing neurons, main arm firing neurons and place preference firing neurons. **b**, **f** The mean and SEM of Z-score of inhibited cells that are responsive to the choice phase in LCA3 or RCA3. **c**, **g** The firing heatmap of place preferred (left or right goal arm) neurons in LCA3 or RCA3. Heatmap rows represent the Z-score-transformed average peri-event histogram for individual neurons when mice turn left or right to the goal arm. Plots to the right showed the mean and SEM of Z-score responses for place preference excited cells. **d**, **h** Prediction firing neurons and main arm firing neurons in LCA3 or RCA3. Heatmap rows represent the Z-score-transformed average peri-event histogram for individual neurons. Plots to the right showed the mean and SEM of Z-score responses for prediction firing neurons and main arm firing cells. **i** The time of firing initiation in each of the prediction and main arm neurons between the LCA3 and RCA3. LCA3 neurons ($n = 105$ cells) tended to fire earlier before RCA3 neurons (n = 100 cells) (unpaired two-tailed Mann–Whitney U test, LCA3 neurons vs. RCA3 neurons, ****$P < 0.0001$). Data are presented as mean ± SEM. **j** The proportion of prediction firing neurons and main arm firing neurons between LCA3 and RCA3. LCA3 showed higher proportion of prediction firing neurons than RCA3 (two-sided Pearson's chi-squared test, ****$P < 0.0001$). Source data are provided as a Source Data file.

Overall, 70.9% of recorded cells were accounted for the corresponsive cells in LCA3 (62.56%, 142 of 227 units are increased; 8.37%, 19 of 227 units decreased), and similarly, 70.7% were corresponsive cells in RCA3 (65.64%, 128 of 197 units are increased; 5.13%, 10 of 197 units decreased). The non-responsive cells barely differed in LCA3 or RCA3 (Fig. 2a and e). In particular, the recorded neurons with increased activity during the choice phase were divided into three groups based on the characterization of firing time: prediction units (firing before mice initiated the choice phase), main arm units (firing when mice run in the main arm),

and location preference units (firing when mice turned into the goal arm) (Supplementary 2c–e). LCA3 contained approximately two-folds of active neurons than RCA3 during the "prediction" behavioral epoch (46.48% in the left, 66 of 142 units; 25.78% in the right, 33 of 128 units). On the contrary, RCA3 had more active neurons during "entering the main arm" behavioral epoch (27.46% in the left, 39 of 142 units; 52.34% in the right, 67 of 128 units) (Fig. 2d, h and j) in comparison to LCA3. Neurons with decreased activity were slightly higher in LCA3 (8.37%, 19 of 227 units) than that in RCA3 (5.13%, 10 of 197 units) (Fig. 2a–b, e–f). We further

analyzed the firing consistency of those recorded neurons (26.06%, 37 of 142 units in LCA3; 21.88%, 28 of 128 units in RCA3) with place preference which are only firing at the target arm. The average firing rates of these place preference cells in LCA3 increased during either "left-direction run" or "right-direction run". This pattern could be also shown in RCA3 (Fig. 2c and g). Most importantly, the time points of firing initiation for prediction and main arm neurons in LCA3 were earlier than that in RCA3 (Fig. 2i). Together the data suggested that asymmetrical distribution of neurons occurs in left-right CA3 and especially LCA3 neurons tend to fire before RCA3 neurons, implicating that LCA3 might be more relevant to SWM.

**Inhibition of LCA3 neurons impairs the retrieval of SWM.** The neural activity of bilateral CA3 is differentially modulated in the T-maze task, especially during the choice phase. Meanwhile, the asymmetric distribution of the left-right CA3 neurons suggested the lateralization of CA3 in SWM. We next tested how this asymmetrical CA3 neural activity functions in different phases of SWM by applying optogenetic system to inhibit unilateral CA3

pyramidal neurons in different phases of the T-maze task (Fig. 3a).

Regarding photo-inhibition of LCA3 pyramidal neurons in the choice phase, well-trained mice hesitated to turn left or right when they reached the T-junction and spent more time on making choices, which led to the disruption of success rates in task performance during the light-on trials. Suppression of LCA3 neural activity in the sample or early/late delay phases (0–5 s/5–10 s inhibition) did not impair task performance (Fig. 3b and Supplementary Fig. 3a). However, suppression of RCA3 pyramidal neurons did not impair task performance at any phase (Fig. 3c).

We also viewed that activation of LCA3 or RCA3 pyramidal neurons in the choice phases impair task performance (Supplementary Fig. 3b and c). It is noted that CA3 has extensive excitatory interconnection between CA3 pyramidal neurons through commissural fiber synapses and the origin of inputs is from contralateral CA3 pyramidal neurons[16,20]. Therefore, we suspected that the activation of RCA3 pyramidal neurons might interfere the intrinsic neural activity of LCA3. To test this hypothesis, we stimulated RCA3 pyramidal neurons while simultaneously recording the

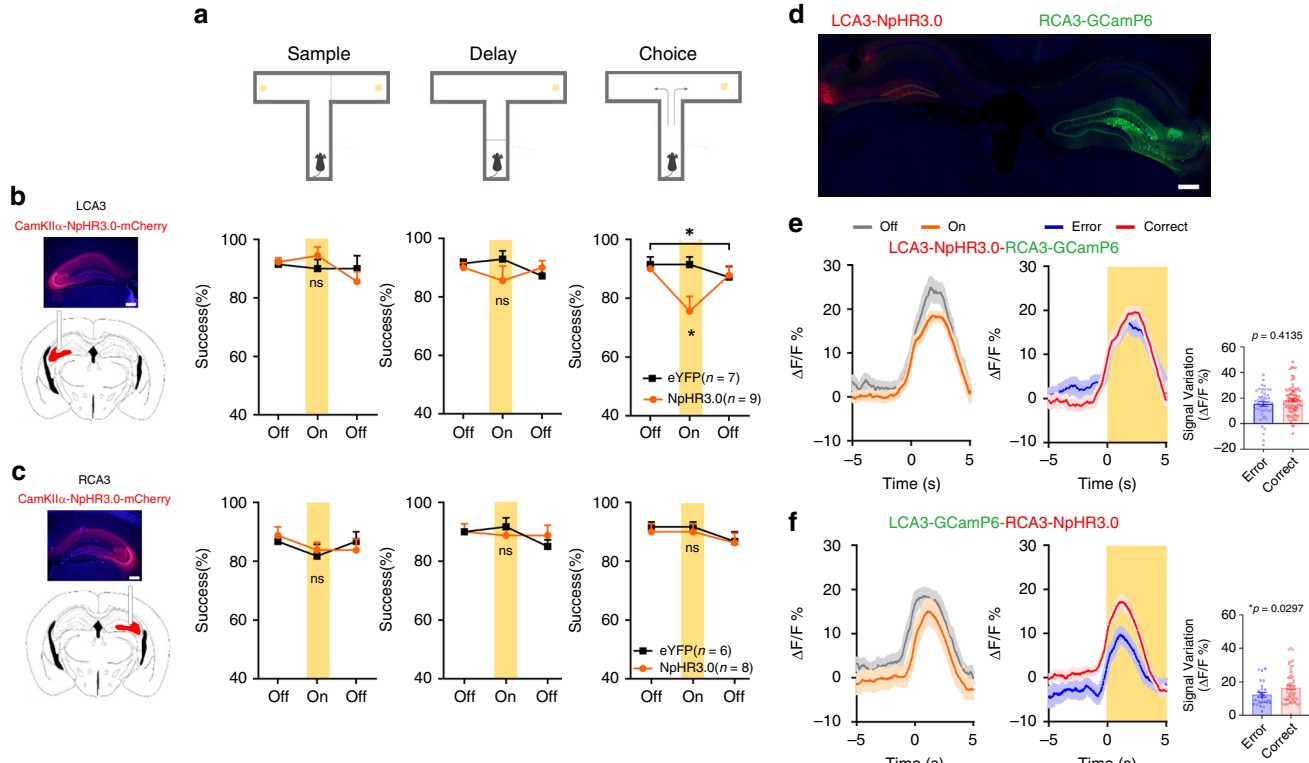

**Fig. 3 Inhibition of LCA3 impairs the SWM in T-maze task. a** Schematic of tDNMTP task. **b, c** Left: viral injection and optic fiber implantation for inhibiting LCA3 neurons (**b**) or RCA3 neurons (**c**). Scale bar: 200 μm. Right: the percentage of correct performance in mice receiving LCA3 ($n = 7$ mCherry, $n = 9$ NpHR3.0) or RCA3 illumination ($n = 6$ mCherry, $n = 8$ NpHR3.0) during the sample phase (a two-way repeated measures ANOVA (rmANOVA), light vs group, LCA3, $P = 0.3476$; RCA3, $P = 0.52$), during the last 5 s of the delay phase (rmANOVA, light vs group, LCA3, $P = 0.21$; RCA3, $P = 0.54$), and during the choice phase (rmANOVA, light vs group, LCA3, *$P = 0.0295$; RCA3, $P = 0.97$). **d** Representative confocal images of NpHR3.0 expression in the LCA3 and GCamp6s expression in the RCA3 (blue, DAPI; green, GCamp6s; red, mCherry). Scale bar: 200 μm. **e** Left: the mean and SEM of Ca$^{2+}$ signal in RCA3 neurons during LCA3 photo-inhibition in the choice phases from five mice. Middle: the mean and SEM of Ca$^{2+}$ signal in RCA3 neurons in choice phase between the correct and error trials during LCA3 pyramidal neurons inhibition. Right: quantification of the average amplitude of increased signal against baseline between the correct and error trials in RCA3 at the choice phase (error: $n = 26$ trials; correct: $n = 65$ trials; unpaired two-tailed Mann–Whitney-U-test, $P = 0.4135$). **f** Left: the mean and SEM of Ca$^{2+}$ signal in LCA3 neurons during inhibition of RCA3 neurons in the choice phases from five mice. Middle: the mean and SEM of Ca$^{2+}$ signal in LCA3 neurons at the choice phase between the correct and error trials during RCA3 pyramidal neurons inhibition. Right: quantification of the average amplitude of increased signals against baseline between the correct and error trials in LCA3 at the choice phase (error: $n = 26$ trials; correct: $n = 65$ trials; unpaired two-tailed Mann–Whitney-U-test, *$P = 0.0297$). Data are presented as mean + SEM (**b**, **c** right panels) and mean ± SEM (**e**, **f** right panels). Source data are provided as a Source Data file.

neural activity of LCA3 pyramidal neurons. RCA3 pyramidal neurons were activated by injecting AAV-CamKIIα-ChrimsonR-tdTomato which was excited by 590 nm yellow light via right optic fiber, and LCA3 pyramidal neurons were recorded by injecting AAV-Syn-GCamP6s that was excited by constant 470 nm blue light via left optic fiber (Supplementary Fig. 3d). For free moving mice, the fluorescence of GCamP6s signal in LCA3 increased sharply along with the activation of RCA3 pyramidal neurons. This outcome implies that the intrinsic neural activity of LCA3 could be disturbed by activation of RCA3 neurons (Supplementary Fig. 3e).

Since PV interneurons project inhibitory inputs to pyramidal neurons, we hypothesized that the activation of PV interneurons could decrease the firing activity of pyramidal neurons to interfere SWM in mice. To test this hypothesis, we employed optogenetics to activate PV neurons and examined the consequences on SWM. The virus of AAV-DIO-ChETA-EYFP was delivered into unilateral CA3 of PV-Cre mice. We observed that optogenetic activation of PV neurons in LCA3, but not in RCA3, resulted in decreased success rates of task performance during the light-on trials (Supplementary Fig. 3f and g), which produced the same consequence as suppression of pyramidal neurons of LCA3. These suggested that activation of PV neurons suppressed CA3 pyramidal neurons and reduced SWM performance.

To evaluate the effects of contralateral inhibition of CA3 neurons on ipsilateral CA3 neural activity, we expressed NpHR3.0 in contralateral CA3 neurons and GCamP6s in ipsilateral CA3 neurons. We connected two optical fibers on bilateral CA3 to deliver 590 nm light for optogenetic inhibition as well as to detect $Ca^{2+}$ signal, respectively. During the T-maze task, we manipulated NpHR3.0 to suppress ipsilateral CA3 activity in the choice phase, which affected contralateral CA3 activity very slightly (Fig. 3e, f). The distinct activity pattern of LCA3 or RCA3 in the choice phase between the correct or error trials was the same as the choice phase light-off trials (Fig. 3e, f; Fig. 1c, d). This evidence indicated that unilateral CA3 neurons have little impact on the firing patterns of the contralateral CA3 neurons during tDNMTP task.

We next investigated whether the lateralization of hippocampal CA3 in SWM could be transferred to other task diagrams by using two different SWM tasks. Applying a delay-match-to-place task in the water maze (wDMTP)[21], mice were trained for 6 days. On day 5, unilateral CA3 pyramidal neurons in the second trial were inhibited, when mice needed to retrieve the memory of the platform location from the first trial (Fig. 4a). Compared with day 4 when mice were well trained, inhibition of LCA3 neural activity on day 5 significantly increased the time of finding platform in the second trial, which remained unchanged under inhibition of RCA3 neurons (Fig. 4a). Meanwhile, on day 6, the behavioral performance of the NpHR3.0-expressed group went back to normal, the same as the control group, which indicated there were no side effects of optogenetic stimulation (Fig. 4b). In addition, during the touchscreen-equipped operant chamber delay-no-match-to-location (toDNMTL) task (Fig. 4c), a delay of 2 s did not show any significant changes in task performance when either LCA3 or RCA3 neurons were optogenetically inhibited. Instead, a delay of 5 s significantly decreased the success rates only when LCA3 neurons were inhibited, rather than RCA3 (Fig. 4d). These results confirmed the critical role of LCA3 neurons on the retrieval of SWM.

## MS^PV-LCA3 projection dominates the lateralization in SWM.
Though the neural activity of bilateral CA3 both increased at similar levels in T-maze task, that of LCA3 but not RCA3 differed between the correct and error trials. We wondered whether bilateral CA3 neurons receive distinct input projections from upstream brain regions that differentially modulate CA3 neural

activity during T-maze task. It has been shown that GABAergic neurons mediating cholinergic neurons in the medial septum and the diagonal band of Broca (MS/DB) have an impact on SWM, while chemical lesions of specific cholinergic neurons in MS leave SWM intact[22–24]. As such, we chose to investigate the neural specificity of dual projections from MS to bilateral CA3.

To investigate the monosynaptic inputs from MS to PV-positive GABAergic and pyramidal neurons in bilateral CA3, we used the rabies virus (RV) -mediated retrograde trans-synaptic tracing system. We first expressed avian-specific retroviral vector TVA containing rabies glycoprotein (RG) specifically in PV or pyramidal cells of CA3. This formula was created by injecting a mixture of two Cre-dependent AAV vectors (AAV-DIO-TVA-BFP and AAV-DIO-RVG) to the bilateral CA3 of CamKII-Cre mice or PV-Cre mice. Three weeks later, the same mice received unilateral injections of the EnVA-pseudotyped RΔG-GFP or EnVA-pseudotyped RΔG into the same region of the bilateral CA3 and were then sacrificed after seven days of RV injection (Fig. 5a). Our results revealed that input neurons are predominantly located in MS. Immunohistochemical analysis was employed to identify the types of input cells in MS. For CamKIIα-Cre mice, 25.9% of the RV-labeled neurons were cholinergic neurons, and only 2.8% of RV-labeled neurons were PV positive (mCherry+white+ or GFP+white+) (Fig. 5b, c, f). This suggested cholinergic and PV neurons in MS form monosynaptic connections with pyramidal neurons in unilateral CA3. There are a few overlapping neurons that were co-labeled with GFP and mCherry (6.15%). Among these overlapping cells, a majority of them were cholinergic neurons (58.33%, mCherry+ GFP+white+) (Fig. 5f), indicating MS cholinergic neurons can also project to pyramidal neurons in bilateral CA3. For PV-Cre mice, 17.4% of RV-labeled neurons in MS/DB were cholinergic neurons and 11.8% were PV-positive neurons (mCherry+white+ or GFP+white+). However, GFP+ and mCherry+ cells were seldom overlapped in MS, suggesting neurons projected from MS to the unilateral CA3 PV neurons form monosynaptic connections and function separately (Fig. 5d–f). These suggested PV neurons in MS/DB mainly form monosynaptic connections with the PV neurons in unilateral CA3.

Previous studies proposed that the cholinergic neurons in MS projecting to the hippocampus form two types of synaptic structures, the formation of typical monosynaptic connections and the dispersed release of acetylcholine to the hippocampus rather than forming synapses[25]. In order to fully understand how MS cholinergic neurons project to CA3, we also applied AAV retrograde tracing system due to the specific property of retroviruses being absorbed by the axon terminus of neurons. AAV-retro-Cre and AAV-retro-Flpo viruses were injected into the bilateral CA3, respectively. In addition, the mixture of AAV-DIO-GFP and AAV-fDIO-tdTomato were injected into MS region. Mice were sacrificed after five weeks of injection (Fig. 5g). Images showed that 23.64% of cells were overlapped, and among these cells, 70.29% were cholinergic neurons (Fig. 5h, i and j), implying that though a portion of MS cholinergic neurons can project to bilateral CA3, they do not form synaptic structures. Comprehensively, these data approved the monosynaptic connections with unilateral CA3 neurons are mainly projected from MS^PV neurons, rather than MS cholinergic neurons, which suggested that MS^PV neurons may play a crucial role in CA3 lateralization in SWM.

## The lateralization of MS^PV-CA3 neurons is SWM task dependent.
To examine the functionality of the projection of MS^PV/Chat → CA3 neurons, we applied in vivo recordings of bilateral CA3 neurons during activation of MS^PV/Chat neurons by expressing AAV-DIO-ChETA-EYFP in MS of PV-Cre mice or Chat-Cre mice

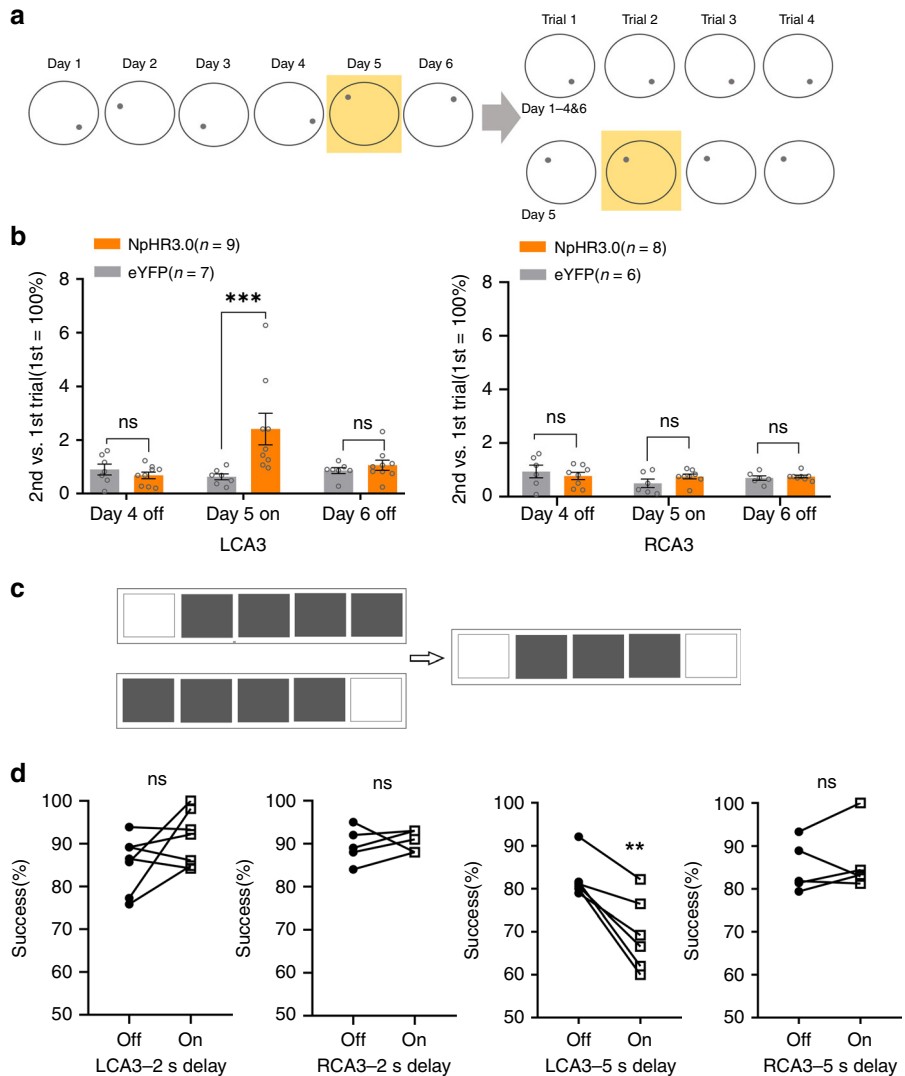

**Fig. 4 Inhibition of LCA3 impairs the SWM in different tasks. a** Schematic of water maze delay match-to-place task (wDMTP). **b** The wDMTP task performance on day 4, 5, 6. Left: LCA3 inhibition was done in the second trial of day 5. wDMTP performance ($n = 7$ eYFP (black), $n = 9$ NpHR3.0 (orange)) on day 4 to day 6 (Bonferroni's multiple comparisons test, day 4: $P > 0.9999$; day 5: ***$P = 0.0006$; day 6: $P > 0.9999$). Right: RCA3 inhibition was done in the second trial of day 5. wDMTP performance ($n = 7$ eYFP (black), $n = 9$ NpHR3.0 (orange)) on day 4 to day 6 (Bonferroni's multiple comparisons test, day 4: $P = 0.7556$; day 5: $P = 0.4449$; day 6: $P = 0.9863$). **c** Schematic representation of the touchscreen-equipped Operant chamber for delayed no-match-to-location task (toDNMTL). **d** The toDNMTL performance with 2 s and 5 s delay (2 s delay: two-tailed paired t-test, LCA3: $P = 0.1394$, RCA3: $P = 0.6548$; 5 s delay: two-tailed paired t-test, LCA3: **$P = 0.0033$, RCA3: $P = 0.5520$). Data are presented as mean ± SEM (**b**, **d**). Source data are provided as a Source Data file.

(Supplementary Fig. 4a, d). In PV-Cre mice, we recorded a proportion of neurons with increased activities in LCA3 (38.46%, 5 of 13 units) and in RCA3 (24.25%, 8 of 33 units) by 20 Hz stimulation of PV neurons in MS (Supplementary Fig. 4b, c). Similarly, in Chat-Cre mice, neurons with increased activities in LCA3 (10%, 2 of 20 units) and RCA3 (9.09%, 2 of 22 units) were also recorded, though in less numbers and with lower firing rates than that in PV-Cre mice (Supplementary Fig. 4e, f). These data implicated the functional connectivity of the projections from $MS^{PV/Chat}$ to CA3 neurons.

To reveal the left-right lateralization of CA3 function on SWM, we explored the influences of $MS^{PV/Chat} \rightarrow CA3$ pathway on SWM through optogenetic inhibition. We expressed NpHR3.0 in $MS^{PV/Chat}$-LCA3 (or RCA3) projecting neurons by injecting AAVretro-Flex-Flpo into LCA3 or RCA3 and AAV-fDIO-NpHR3.0-mcherry into MS of PV-Cre mice (Fig. 6a) or

Chat-Cre mice (Fig. 6d). The immunofluorescent staining showed that there were co-localized cells (mCherry$^+$white$^+$) in both PV-Cre and Chat-Cre mice, proving that the projection neurons from MS were PV-positive or Chat positive neurons (Fig. 6b, e). We inhibited NpHR3.0 labeled neurons in the choice phase through applying 590 nm laser in the experimental group (or 470 nm in control group) and found that the PV-Cre mice showed impaired performance of the T-maze task in $MS^{PV}$-LCA3 group, while there were no obvious changes of task performance in the $MS^{PV}$-RCA3 group (Fig. 6c). However, the Chat-Cre mice did not show significant differences of success rates of T-maze task when either $MS^{Chat}$-LCA3 or $MS^{Chat}$-RCA3 was inhibited in the choice phase (Fig. 6f). These suggested that $MS^{PV}$-CA3 neurons, rather than $MS^{Chat}$-CA3 neurons, that are distributed unilaterally to CA3, mainly contribute to the LCA3 dominance in the choice phase of SWM.

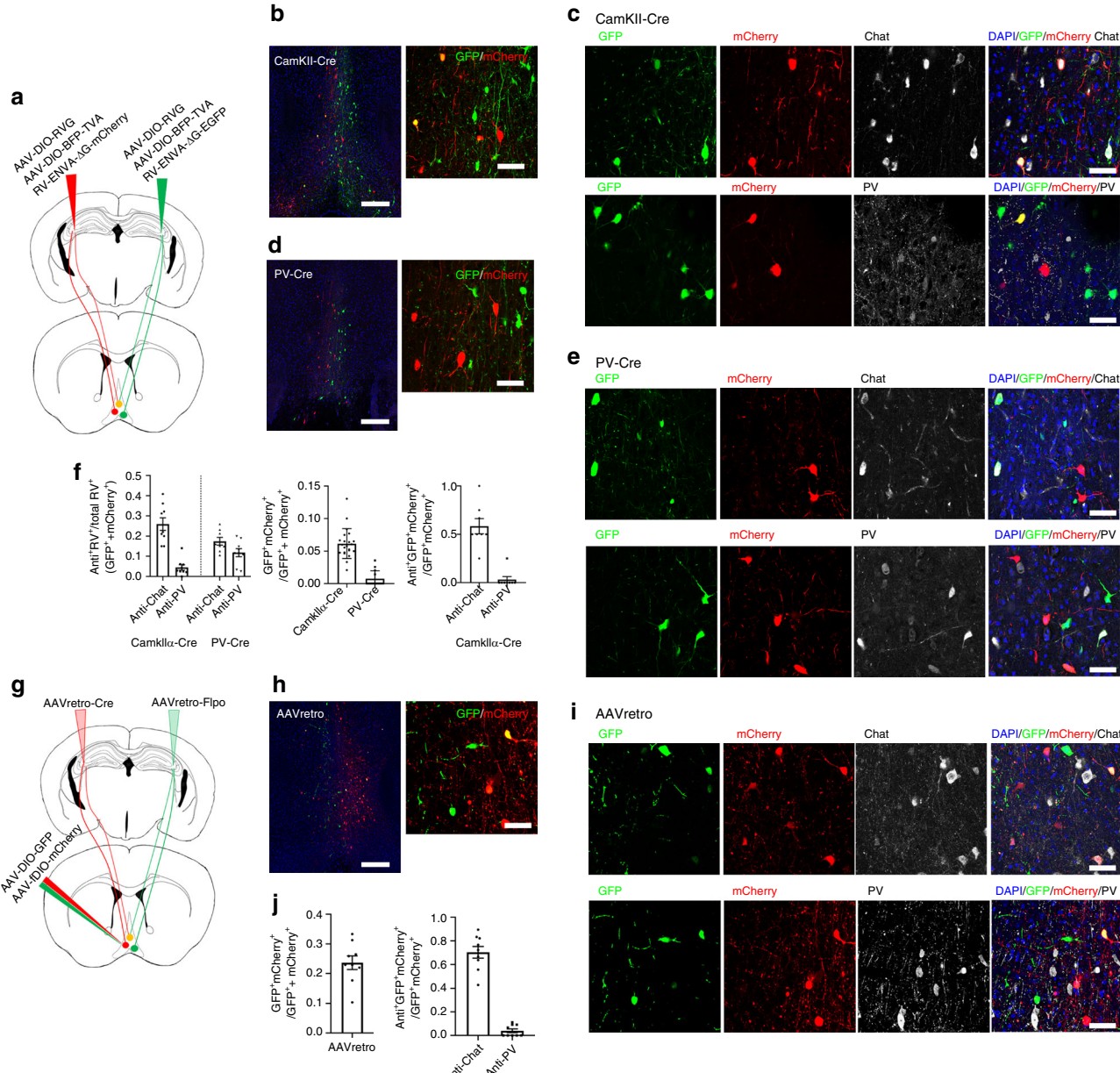

**Fig. 5 MS^PV neurons form monosynaptic connections with unilateral CA3. a** Schematic representation of monosynaptic rabies retrograde tracing from bilateral CA3 pyramidal neurons in CamKIIα-Cre mice or PV neurons in PV-Cre mice. **b**, **c** Confocal images of coronal sections showing RV input neurons (GFP+ or mCherry+) in MS from the tracing brains of CamKIIα-Cre mice were co-labeled with cholinergic (white, up) or PV (white, bottom) neurons. Scale bar: left is 200 μm; right is 50 μm. **d**, **e** Confocal images of coronal sections showing RV input neurons (GFP+ or mCherry+) in MS from the tracing brains of PV-Cre mice were co-labeled with cholinergic (white+, up) or PV (white, bottom) neurons; Scale bar: left is 200 μm, right is 50 μm. **f** Left: quantification of percentage of all RV labeling cells (GFP+ or mCherry+) that were cholinergic neurons or PV neurons in CamkIIα-Cre ($n = 9$ images from three mice) or PV-Cre mice ($n = 9$ images from three mice). Middle: quantification of percentage of all RV labeling cells were double-positive neurons (GFP+mCherry+) in CamkIIα-Cre or PV-Cre mice. Right: quantification of percentage of double-positive cells that were cholinergic neurons or PV neurons in CamkIIα-Cre mice. **g** Schematic of AAV-retro tracing system. Injection of AAVretro-Flpo and AAVretro-Cre in bilateral CA3 respectively and a mixture of AAV-DIO-GFP and AAV-fDIO-tdTomato in MS. **h**, **i** Confocal images of coronal sections showing Cre-positive neurons (GFP+) and Flpo-positive neurons (mCherry+) in MS were co-labeled with cholinergic (white, up) or PV (white, bottom) neurons. Scale bar: left is 200 μm, right is 50 μm. **j** Left: quantification of the percentage of AAVretro labeling cells that were double-positive neurons (GFP+mCherry+). Right: quantification of the percentage of double AAVretro labeling cells (GFP+ and mCherry+) that were cholinergic neurons or PV neurons in WT mice ($n = 9$ images from three mice). Data are presented as mean ± SEM (**f**, **g**). Source data are provided as a Source Data file.

## Discussion

Hippocampal CA3 has been widely known to involve in memory encoding and recall within second-long intervals, and be well adapted for the rapid storage and retrieval of associative memories.

Dorsal CA3 has been reported to affect the tDNMTP task via Fimbria from MS through permanent lesions[26,27]. However, the different phases of SWM in eight-arm radial maze may not be discriminated well. We found that CA3 neuronal firing rates were

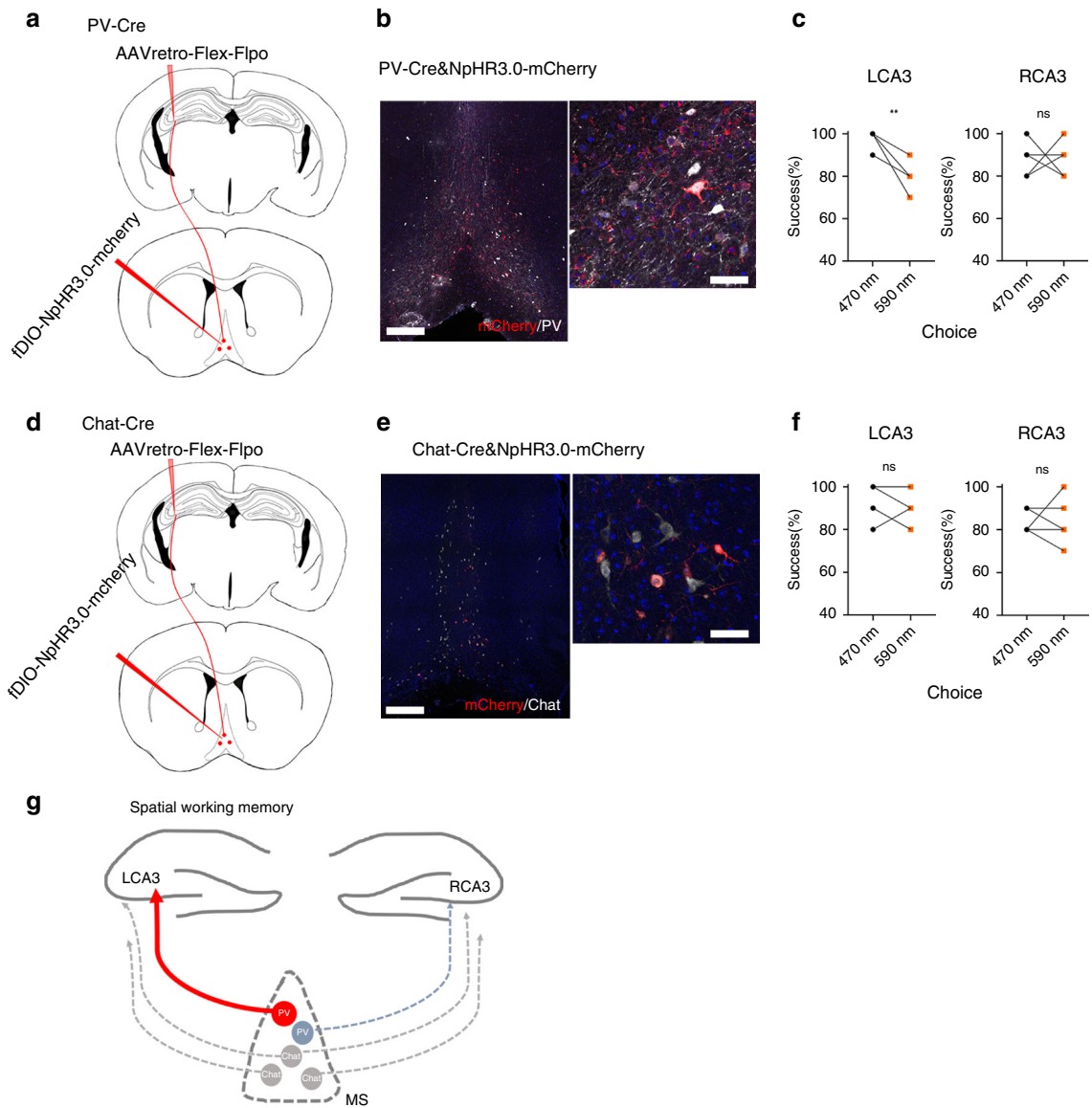

**Fig. 6 Inhibition of MS$^{PV}$-LCA3 projection impairs the retrieval of SWM. a** Schematic of AAV-retro retrograde optogenetic system in PV-Cre mice. **b** Confocal images of coronal sections showing that mCherry$^+$ cells were co-labeled with PV (white) neurons. Scale bar: left is 200 μm, right is 50 μm. **c** Percentage of correct performance in mice receiving inhibition of MS$^{PV}$-LCA3 ($n = 7$) or MS$^{PV}$-RCA3 ($n = 5$) during the choice phase (two-tailed paired t-test, LCA3: **$P = 0.0019$, RCA3: $P > 0.9999$). **d** Schematic of AAV-retro retrograde optogenetic system in Chat-Cre mice. **e** Confocal images of coronal sections showing that mCherry$^+$ cells were co-labeled with cholinergic (white) neurons. Scale bar: left is 200 μm, right is 50 μm. **f** Percentage of correct performance in mice receiving inhibition of MS$^{Chat}$-LCA3 ($n = 6$) or MS$^{Chat}$-RCA3 ($n = 6$) during the choice phase (two-tailed paired t-test, LCA3: $P = 0.2031$, RCA3: $P = 0.7412$). **g** A proposed model for MS$^{PV}$-LCA3 projection that modulates the lateralization of SWM. Source data are provided as a Source Data file.

increased when the choice phase was initiated, and remained unchanged when mice reached the reward at the end of the goal arm. Inhibition of LCA3 pyramidal neurons (or activation of CA3 PV neurons) in the choice phase impaired SWM performance, while this suppression in the sample or delay phase kept SWM intact. This suggests that LCA3 dominates the retrieval of SWM.

It has been evidenced that LCA3 and RCA3 in CA1 show different postsynaptic projections[7]. Only LCA3 presynaptic input induced high-frequency stimulation could increase LTP[8]. However, it is still unclear that how the neural firing patterns in LCA3 or RCA3 mediate SWM. Moreover, the possible projection pathways between CA3 and other brain regions in regulating SWM remain unknown. Our data indicated that the subpopulation of CA3 neurons expresses different neural activity between

LCA3 and RCA3 in the choice phase of SWM. Though similar fractions of neural units recorded in LCA3 and RCA3 were found to be responsive to the choice phase during tasks, LCA3 contains more active prediction neurons than RCA3 and LCA3 neurons tended to initiate firing much earlier than RCA3 neurons. These data implicated LCA3 might play an essential role in the choice phase of SWM.

It has been proposed that inhibitory neurons (mostly interneurons, INs) interact with each other, form neural circuitry of inhibitory synaptic connections and also reciprocally connect with and regulate other excitatory neurons to manipulate learning and memory[28–30]. Activation of PV-positive interneurons was found to be highly effective in suppressing pyramidal neuronal activity[31,32]. Dan team claimed that suppression of pyramidal

neuronal activity by activation of PV-positive interneurons could impair the memory-guided task performance[33]. In CA3, the neuronal excitability, firing and synaptic integration of its pyramidal neurons can be regulated by receiving strong and diverse GABAergic inputs, such as PV-positive cells, PV-containing axo-axonic cells, which has been reported to be able to regulate the spiking of CA3 pyramidal neurons[16,34]. This is a strong support for our selection of activating PV interneurons as a choice for suppressing pyramidal neurons in LCA3. As expected, the SWM was affected by activation of PV interneurons similarly as that by only inhibition of LCA3 pyramidal cells.

It is well known that CA3 plays specific roles in hippocampal function and memory, especially for the sparse inputs from the DG to CA3 and for the extended local recurrent connectivity that operates the CA3 forming extensive excitatory interconnections between CA3 pyramidal cells through associative and commissural (A/C) fiber synapses[16]. Our data suggested inhibition of LCA3 pyramidal neurons, rather than RCA3, could reduce the success rates of mice in the choice phase of tDNMTP tasks. However, activation of either LCA3 or RCA3 pyramidal neurons in the choice phase impairs the task performances. We speculated that the activation of RCA3 pyramidal neurons might interfere the intrinsic neural activity of LCA3 and this was confirmed by the results that the GCaMP6s signal in LCA3 increased sharply along with the activation of RCA3 pyramidal neurons. We also performed experiments to exclude the effects of contralateral inhibition of CA3 neurons on ipsilateral CA3 neural activity. The data suggested that suppression of the ipsilateral CA3 neuronal activity in the choice phase seldom affects the contralateral CA3 neuronal activity, strongly implicating the critical role of LCA3 neurons on the retrieval of SWM.

Though application of optogenetics on neuroscience has been widely accepted recent years, it is not deniable that some disadvantages of its use, such as, the intensity of laser or the time duration of stimulation, need to be concerned. It was reported that higher intensive laser or prolonged stimulation by laser light can produce heat to brain tissue and thereby lead to inhibition or damages of the neuronal activity as well as behavioral ability in the absence of opsins[35,36]. Regarding this, we limited the time durations of laser stimulation between 3 and 5 s, which could not only cover the initiation of the choice phase (which means it can cover the time points of firing of those prediction and main arm neurons), but also avoid the side effects caused by the high intensity and long-lasted stimulation of laser light. In our optogenetic experiments, light was turned on at the same time as the door was open during the sample and the choice run, so 590/470 nm of laser manipulation could cover the beginning of the initiation of the choice run. In particular, 590 nm of laser was lasted for 5 s for the inhibition of CA3 neurons, while 470 nm of laser was lasted for 3 s for activation of CA3 neurons, avoiding the epilepsy seizure of mice by 5 s activation. Nevertheless, the GtACR1, a new tool of silencing neurons by optogenetics has been recently created, can be activated by much powerless laser to inhibit targeted neurons than NpHR3.0[37]. It would be considered for inhibiting neurons in future studies.

Retrograde trans-synaptic rabies virus system is prevalently applied in brain-wide maps on cell-type-dependent global scale input patterns. As one of the upstream projecting regions to the hippocampus, the MS/DB has been emerged as a key modulator for hippocampal function via septo-hippocampal pathway. Both GABAergic and cholinergic neurons in MS/DB are correlated with learning and memory and hippocampal rhythmogenesis[22,38]. Earlier studies have demonstrated that the septal cholinergic neurons project diffusely to both the interneurons and pyramidal neurons in hippocampus, which is contrast with the GABAergic MS/DB neurons that exclusively innervate hippocampal interneurons[39,40]. Dannenberg H team

proposed that PV neurons in MS/DB would be recruited by cholinergic neurons to selectively innervate hippocampal interneurons and be responsible for precise synchronization of hippocampal networks[41]. It was found that GABAergic lesions of the MS/DB in rats could impair the hippocampal acetylcholine efflux and the spatial working memory in the tDNMTP tasks[22]. Recent study also demonstrated that optogenetic stimulation of MS-PV neurons could recover slow gamma oscillations in hippocampus and rescue the spatial memory of AD model mice[42]. In our data, we mapped the anatomical specificity and organization of MS to unilateral CA3 with distinct neuronal types. We revealed that MS cholinergic neurons mainly project to bilateral CA3 pyramidal neurons, while PV neurons in MS project to unilateral CA3 neurons. Selective suppression of MS[PV]-LCA3 projecting neurons impaired the performance of SWM during the choice phase, suggesting that PV neurons in MS have potential to dominate the lateralization of LCA3 in SWM. Regarding the global effects of MS GABAergic inputs to hippocampus, whether MS GABAergic inputs function similarly to regulate hippocampal CA3 via manipulating its oscillatory patterns, or whether the influence on the process of memory was permissive, should be addressed. Further exploration would focus on revealing whether the lateralization of LCA3 would be associated with the oscillatory modulation of MS GABAergic inputs to hippocampal CA3 and what types of CA3 neurons would be involved in this modulation.

Our findings also revealed that the MS[PV]-LCA3 projection dominates SWM task, but does not show remarkable effects compared with MS[PV]-RCA3 without performing task, suggesting the lateralization of LCA3 is task dependent. Actually, some studies have reported similar findings that the left prioritization usually occurs along with tasks. In human behavioral research, not only hippocampus, but also dorsolateral prefrontal cortex show lateralization, which are tasks related, such as spatial memory and navigational tasks[5,43].

It was reported that the β2 oscillations (23–23 Hz) in CA3 showed strong bursts when mice explore novel, rather than familiar, environments; the theta (10 Hz) and beta (20 Hz) band synchronization of neuronal firing patterns and local field potential activity in mice CA1 increased in the reward cue tasks[38,44]. To detect the neuronal firing changes, we selected 20 Hz for stimulation of MS neurons and recorded neurons with increased activities in CA3 in both PV-Cre and Chat-Cre mice, suggesting the projections from MS[PV/Chat] to CA3 are functionally connected.

Overall, our results indicate that the lateralization of LCA3 plays a vital role in SWM, especially during memory retrieval. The PV and cholinergic neurons in MS have different projection patterns to CA3. MS[PV] neurons tend to project to unilateral CA3 and form functional connections, which affects the behavior of SWM in mice. Our data provides an insight into the lateralization of LCA3 during the retrieval of SWM.

## Methods

**Approvals**. All surgical and experimental procedures were approved by the Institutional Animal Care and Use Committee of the Beijing Institute of Technology, Beijing, China.

**Subjects**. Adult (2–4-month-old) male and female WT (WT, C57/BL6), PV-Cre (Jax Stock no. 008069), CamKIIα-Cre (Jax Stock no. 013044) and Chat-Cre mice (Jax Stock no. 006140) mice were used. Mice were group-housed on a 12/12 h light/dark cycle (2–5 animals per cage) at a consistent ambient temperature (23 ± 1 °C) and humidity (50 ± 5%), and all experiments were performed during the light cycle. Food and water were accessed ad libitum, except when on food restriction during behavioral tests. Littermates were randomly assigned to each condition by the experimenter. Only male mice were used in the behavioral tests.

**Viruses**. For rabies virus-mediated retrograde tracing, AAV9- Ef1α-DIO-RVG (titer, $5.1 \times 10^{12}$ genome copies (gc) per ml), AAV9-Ef1α-DIO-BFP-TVA (titer, $5.1 \times 10^{12}$ gc ml$^{-1}$), and EnvA-pseudotyped, glycoprotein (RG)-deleted rabies virus

RV-EvnA-GFP (titer, $6.50 \times 10^8$ colony forming units (cfu) per ml) and RV-EvnA-mCherry (titer, $5.00 \times 10^8$ cfu ml$^{-1}$) were purchased from Brain VTA, Wuhan, China.

AAVs particles (AAV9-Syn-GCamP6s, titer, $3.5 \times 10^{12}$ gc ml$^{-1}$; AAV9-CamKIIα-ChR2-EYFP, titer, $5.1 \times 10^{12}$ gc ml$^{-1}$; AAV9-CamKIIα-EYFP, titer, $4.2 \times 10^{12}$ gc ml$^{-1}$; AAV9-CamKIIα-NpHR3.0-mCherry, titer, $5.1 \times 10^{12}$ gc ml$^{-1}$; AAV8-CamKIIα-ChrimsonR-mCherry, titer, $2.8 \times 10^{12}$ gc ml$^{-1}$; AAVretro-Syn-Cre, titer, $1.5 \times 10^{13}$ gc ml$^{-1}$; AAVretro-Syn-Flpo, titer, $1.2 \times 10^{13}$ gc ml$^{-1}$; AAV9-Ef1α-DIO-GFP, titer, $3.0 \times 10^{12}$ gc ml$^{-1}$; AAV9-Ef1α-fDIO-tdTomato, titer, $6.2 \times 10^{12}$ gc ml$^{-1}$; AAV9-Ef1α-DIO-ChETA-EYFP, titer, $4.2 \times 10^{12}$ gc ml$^{-1}$) were purchased from Taitool, Shanghai, China.

**Stereotaxic injections.** The animals were deeply anesthetized and placed in a stereotactic frame (RWD, Shenzhen, China). Ophthalmic ointment was applied to prevent dehydration. The virus was injected by a 10 μL Hamilton microsyringe at a constant speed with a microsyringe pump (UMP3; WPI, Sarasota, FL, USA) and controller (Micro4; WPI, Sarasota, FL, USA). After viral injection was completed, the needle was held still for 10 min to allow the diffusion of the virus. The needle was then withdrawn slowly and completely.

Measurement of optic fiber transmissivity: we measured light intensity at the laser launch, tip of the coupled fiber (diameter, 200 μm; Ferrule O.D. 1.25 mm; N.A., 0.37; length, 3.0 or 4.0 mm; Inper Inc., China), and the fiber ferrule implant by optical power meter (PM121D, thorlab, USA). Fiber ferules were sorted based on optical transmissivity (>80%). We measured intensity at the coupled fiber before connecting to the implanted fiber ferrule and connecting to each animal every day. The intensity was kept to adjust to certain power (ChR2/ChETA simulation: 1 mW; NpHR3.0 stimulation:10 mW; fiber photometry: 40 μW).

For fiber photometry recordings, 300 nl of AAV-Syn-GCamP6s was injected into the left or right hemisphere of CA3 region (anteroposterior (AP): −2.06 mm; mediolateral (ML): ± 2.35 mm; dorsoventral (DV): −2.35 mm). Also, Fiber ferrule was placed 50 μm above the viral injection site. The optical fiber was secured to the skull using jeweler's screws and dental cement.

For optogenetic stimulation of CA3 neurons experiments, 300 nl of AAV2/8-CamKIIα-eNpHR3.0-EYFP, 300 nl of AAV2/8-CamKIIα-ChR2 (H134R)-EYFP or 300 nl of AAV2/8-CamKIIα-EYFP was injected into left or right hemisphere of CA3 region (AP: −2.06 mm; ML: ± 2.35 mm; DV: −2.35 mm). The fiber ferrule was placed 100 μm above the virus injection site and secured with dental cement and jeweler's screws.

For fiber photometry recoding and optogenetic stimulation in contralateral CA3 concurrently, 300 nl of AAV2/9-CamKIIα-NpHR3.0-mcherry (or AAV2/8-CamKIIα-ChrimsonR-mCherry) and 300 nl of AAV5-Syn-GCamp6s was injected into left or right hemisphere of CA3 (AP: −2.06 mm; ML: ± 2.35 mm; DV: −2.35 mm). The optical fiber was placed 50 or 100 μm above the virus injection site and secured with jeweler's screws and dental cement.

For rabies tracing experiments, AAV9-Ef1α-DIO-BFP-TVA and AAV9- Ef1α-DIO-RVG (volume ratio: 1:1, the total volume of 400 nl) were injected into the dual CA3 regions of CamkIIα-Cre and PV-Cre mice at the following coordinates: AP: −2.06 mm; ML: ± 2.35 mm; DV: −2.35 mm. Four weeks later, RV-EnVA-EGFP and RV-EnVA-mCherry (volume, 350 nl) were injected into the unilateral CA3 separately. Mice were sacrificed 7 days after rabies virus infection.

For AAVretro tracing experiments, AAVretro-Syn-Cre and AAVretro-Syn-Flpo (each volume of 300 nl) were injected into the bilateral CA3 regions at the following coordinates: AP: −2.06 mm; ML: ± 2.35 mm; DV: −2.35 mm. AAV9-Ef1α-DIO-GFP and AAV9-Ef1α-fDIO-tdTomato (volume ratio: 1:1, total volume of 400 nl) were injected into MS at the following coordinates: AP: 0.8 mm; ML: 0 mm; DV: −3.8 mm.

For in vivo neurophysiological experiments, mice were implanted with a moveable microdriver consisting of a 32-channel electronic interface board. A total of eight tetrodes was assembled and mounted onto the microdriver. A tetrode consists of four 25 mm platinum (with 10% iridium) wires (California Fine Wire, USA). The tetrode bundle was then targeted to the CA3 region.

For MS$^{PV}$-CA3 or MS$^{Chat}$-CA3 recording, 200 nl of AAV2/9-Ef1α-DIO-ChETA-EYFP was injected into MS (AP: 0.86 mm; ML: ± 0.2 mm; DV: −3.5 mm) in PV-Cre or Chat-Cre mice. Tetrodes were targeted to CA3 region.

**Immunohistochemistry.** Mice were deeply anesthetized and transcardially perfused with 0.9% saline followed by 4% paraformaldehyde (PFA) in PBS. Brains were extracted, removed and kept in 4% PFA for at least 24 h. Next, brains were transferred to 30% sucrose dissolved in PBS until they sank to the bottom of the container and were then sliced into 50 μm coronal sections using a freezing microtome (Leica, CM3050 S, Germany). The sections were stored at −20 °C in PBS containing 30% glycerol (v per v), 30% ethylene glycol (v per v) until they were processed.

For immunofluorescence staining, free-floating sections were washed with PBS three times (5 min each) and incubated with blocking buffer that contained 5% donkey serum dissolved in 0.3% PBST (0.3% Triton X-100 in PBS) for 2 h. Sections were then incubated with primary antibodies diluted in blocking buffer (2.5% donkey serum) overnight at 4 °C. After incubation, the sections were washed three times (5 min each) with PBST and incubated with a fluorescent dye-conjugated secondary antibody (1:500, Abcam, UK) for 2 h at room temperature. Following

three washes (5 min each time) with PBS, sections were mounted under coverslips. Primary antibodies used were: anti-Chat (1:1000, goat, AB144P, Millipore, USA), anti-PV (1:2000, rabbit, PV27, Swant, USA), anti-2A peptide antibody (1:1000, mouse NBP2-59627SS, Novus Biologicals, USA). Images were acquired and analyzed with a NIS-Elements AR confocal microscope and Image J.

**Fiber photometry recording.** An apparatus for performing fiber photometry recording was obtained from the Thinker Tech Nanjing Biotech Limited Co., Nanjing, China. The signal was digitized and collected by ThorCam-DAQ software. After three weeks of GCamP6s virus expression, calcium signal was recorded when the mouse performed tDNMTP. The timing of behavioral variables was recorded by the same system. The behavioral processes and calcium signal were synchronized and analyzed by the custom-written Matlab software (The MathWorks, Inc., USA).

For calculating GCamP6s signal, relative fluorescence changes of ΔF/F were calculated as Ca$^{2+}$ signal as follows:

$$\Delta F/F = (F_{raw} - F_{baseline})/F_{baseline}, \quad (1)$$

where the $F_{baseline}$ was the baseline fluorescence taken when mice stay in the home cage for 30 s before the T-maze task.

Signal variation of GCamP6s in choice phases was calculated as follows:

$$\text{Signal Variation} = \text{Mean Signal}(1,\ 2) - \text{Mean Signal}(-5, -1). \quad (2)$$

Mean signal (1, 2) stands for the mean increased signal in the time course from 1 s to 2 s, and Mean signal (−5, −1) stands for the mean signal of baseline in the time course from −5 s to −1 s.

**In vivo single-unit recording.** Single unite data was captured across animals (3 mice for LCA3 recording, 3 mice for RCA3 recording). After surgery, mice were allowed for recovery at least one week before recording. Multichannel extracellular signals were recorded from the bilateral hippocampal CA3. Movable microdrivers were advanced around 70 μm daily until we could record the maximal units from target brain area. Signals were amplified, band-pass filtered (0.5–1000 Hz for LFPs and 0.6–6 kHz for spikes) and digitized using the Plexon OmniPlex Neural Data Acquisition System (Dallas, TX, USA). Spikes, detected at adjustable online thresholds, were collected at 40 kHz. Single units were clustered in Offline Sorter and MClust 4.4 program (A. D. Redish, http://redishlab.neuroscience.umn.edu/MClust/MClust.html).

**Single-unit recording analysis.** A peri-stimulus time histogram (PSTH) for the time points of the sample or choice begin was calculated with 50 ms between −5 s and 5 s relative to the sample or choice phases for each neuron. From this, the mean μ and standard deviation σ of the firing rate (−5 s to 5 s) were calculated and used to generate the Z-score normalization:

$$Z - \text{score} = \frac{\text{PSTH} - \mu}{\sigma}. \quad (3)$$

Place preference neurons were determined from 1 s after goal arrival on all left versus right trials in the sample or choice phases. The z-scored firing rate within 1 s of sample phase on all left-versus-right trials was used to determine the significance of L/R CA3 neurons (Wilcoxon's rank-sum test, $P < 0.005$).

Increased activity was determined from z-scored firing rates calculated in 50 ms bins. If a single neuron exhibited a z-scored firing rate beyond 2 s.d. for four or more consecutive bins from −1 s to 2 s, it was classified as an increased neuron. The time point of first bin which is beyond 2 s.d. used to classify the prediction neurons (from −1 s to 0 s) or main arm neurons (from 0 s to 2 s).

Decreased activity was determined from z-scored firing rates calculated in 50 ms bins. If a single neuron exhibited a z-scored firing rate beyond −2 s.d. for four or more consecutive bins from −1 s to 2 s, it was classified as a decreased neuron.

**MS$^{PV}$-CA3 and MS$^{Chat}$-CA3 recording.** Increased/decreased neurons: the averaged firing rate was significantly increased/decreased between baseline window of 1 s prior to 470 nm laser onset and response window of 1 s during laser delivery. The time window for Z-scored calculation is 100 ms (Wilcoxon's rank-sum test, $P < 0.05$).

**T-maze for delayed no-match-to-place (tDNMTP) task.** tDNMTP task was performed to test spatial working memory by manual operation. Mice were gradually food restricted until they reached 85% body weight. Mice were habituated to the T-maze over two days, during which mice were placed into the T-maze for 10 min. 50 μl of 50% sweet-milk reward was placed in the reward tube at the end of choice arms. Once experiments began, mice were given ten trials per day. Each trial was consisted of the sample run, delay and choice run.

On the sample run, the mouse was forced to enter the left or right arm to consume the sweet-milk reward, while the other arm was blocked by a door. Then the mouse came back in the start position. On the choice run, the blocked door was removed and the mouse was allowed to choose one of the goal arms. The time interval between the sample and choice run was 10 s. If the mouse visited the non-sample arm, it was allowed to acquire the reward. If the animal visited sample arm,

it could not receive the reward. If the mouse was consumed the reward, this was scored as a correct trial. The time interval from one trial to the next was about 1 min. After rewards, the mice were allowed to return to the start point by themselves, however, if the mice could not return to the start point by themselves 15 s after feeding, they will be forced to return to the start point. Mice were trained for 10 trials per day until the performance of success at 90% in two consecutive days and then performed opto-stimulating process. On each day, mice had five starts from the left of the sample arm runs and five starts from the right in a pseudorandom order with no more than three consecutive starts from the left or right.

**Touchscreen for delayed no-match-to-location (toDNMTL)task**. A trial of this task consists of three phases, the sample, delay and choice phases. The sample phase was initiated by starting one stimulus that displays (a white square) in left or right locations on the screen and the stimulus disappeared once the mice had a nose-poke of it (the sample). The mice were allowed to delay for 2 or 5 s. To start the next choice phase, the second initiation procedure was designed to prevent mice from mediating during the delay period by waiting in front of to-be correct or to-be-error location. In the choice phase, two stimuli were designed that one was in the old (sample, incorrect) location and the other in the new (correct) location on the other side. Mice touching to the correct location were rewarded and allowed to entering an inter-trial interval (ITI) for the next trial, but a touch to the incorrect location was treated as a 5 s time-out and then an ITI, either followed by correction trials. Mice were allowed to make correction trials that were followed the same procedures as normal trials, excluding the condition that the same sample and choice locations from the previous incorrect trial were kept until mice made the correct choice. Data was acquired with ABET II Touch 2.20.

**Water maze for delayed match-to-place (wDMTP) task**. The delayed matching-to-place task in water maze test was performed successively in six consecutive days. Briefly, the water maze was 1.2 m in diameter, with a 10-cm-diameter circular escape platform. The water temperature was kept at 22–23 °C. The platform, which was placed 1 cm below the surface of the water, was randomly located in one of the four quadrants every day and not in the same quadrant for 2 consecutive days. Mice were given four trials with an interval of 10 s each day at the same starting position. The location of hidden platform was fixed on the same day. Before experiment began, the platform was located above the surface of the water and marked by an attached flag to provide a cue with four trials for one day. The first and second trials each day were used to evaluate working memory by calculating the percentage of latency of finding the platform during the second trial (second trial/first trial). The third and fourth trials were used to train the animal to learn for the day. Data were acquired with EthoVision XT 12.

**Behavioral tests coupled with optogenetic Manipulation**. For the tDNMTP task, mice were given 470 nm or 590 nm light stimulating after the percentage of success rates reach 90% in two consecutive days. For optogenetic inhibition of CA3 using eNpHR3.0, and 10 mW of constant yellow light was delivered and sustained for 5 s by a 590 nm DPSS laser (10 mW, constant light on). For optogenetic using hChR2 (H134R), 470 nm DPSS laser would generate a train of blue light pulses (1 mW, 20 Hz, 5 ms duration) and sustained for 3 s. All of the light on stages were at the stage of the sample phase, the first 5 s/the last 5 s of the delay phase, and the choice phase, respectively. A completed test lasted three consecutive days and each day included three processes (light off, light on and light off).

For the wDMTP task, mice were stimulated with 470 nm light at the second trial in the day 5. For eNpHR3.0 mice, the 590 nm constant light was sustained for 5 s.

For the toDNMTL task, mice were stimulated with 590 nm light when the choice phase initiated and lasted 5 s.

For the home cage optogenetic simulation coupled with single unite recoding, mice were stimulated with 470 nm light (1 mW, 20 Hz, 5 ms duration), which sustained 1 s.

**Statistical analysis and reproducibility**. A two-way repeated measures ANOVA was used to assess significant interactions of light vs group in behavioral experiments. Bonferroni-corrected P values were used and indicated for multiple comparisons. Two group comparisons were analyzed by unpaired two-tailed t-test. In single-unit recording with nonparametric data, we used Wilcoxon's signed-rank test for paired calculation and Mann-Whitney-U-test in unpaired observations. All of the statistical details of experiments can be found in the figure legends and Supplementary Table 1. Data was presented as means ± standard errors of the means (SEM). Statistical analyses were performed with GraphPad Prism 8.0.

Each experiment has been repeated three times in the manuscript with similar results.

**Reporting summary**. Further information on research design is available in the Nature Research Reporting Summary linked to this article.

## Data availability

All the relevant data supporting the findings of this study are available within the article and its supplementary information files or from the corresponding authors upon request. The source data are provided as a Source Data file. Source data are provided with this paper.

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

## Acknowledgements

This work was supported by the National Key R&D Program of China (Grant No. 2017YFE0117000), the National Key Research and Development Program of China (Grant No. 2018YFC1312302-3, 2018YFC0115400), the National Natural Science Foundation of China (Grant Nos. 81870844, 81671268 & 81701260), the Beijing Municipal Science and Technology Commission (Grant No. Z191100010618004), Xinglin Scholar of Shanghai University of Traditional Chinese Medicine (Grant No. A1-U1820501040222), and Budgeted Research Project of Shanghai University of Traditional Chinese Medicine (Grant No.18LK002). We thank Dr. Xiaohui Zhang and Dr. Yousheng Shu from Beijing Normal University for gifts of transgenic mice and professional advices. We thank Dr. Minmin Luo from National Institute of Biological Sciences for guidance of fiber photometry recording. We also thank Dr. Zhantao Bai from Yanan University for scientific suggestions and experimental equipment. We thank the Biological and Medical Engineering Core Facilities of Beijing Institute of Technology for supporting experimental equipments.

## Author contributions

D.S., ZZ.Q., and H.Q. conceived and designed the studies and wrote the papers. D.S., DH.W., and Z.W. performed optogenetics and single-unit recording. QH.Y., YR.L., and TY.Y. carried out fiber photometry recordings, CJ.W., T.H., T.J., and YL.W. carried out tDNMTP, wDMTP, and toDNMTL behavioral tests; J.Z., YC.L., and HA.Z. performed immunohistochemistry and cell counting. DH.W., ZJ.K., and D. S. performed rabies virus-mediated retrograde tracing experiments. Y.Y., YH.L., Q.S., and Z.X. performed injections and imaging. All authors contributed to the data analysis and presentation in the paper.

## Competing interests

The authors declare no competing interests.
