## [Peer Review File · Nature Communications]

Reviewers' comments:

Reviewer #1 (Remarks to the Author):

In this study, the authors set out to address two questions concerning the activity of hippocampus CA3 during working memory tasks: (1) do bilateral CA3 neurons have different firing patterns in the working memory process, and (2) is any potential difference between left and right CA3 the result of upstream projections from the Medial Septum.

My major issue with the study stems from the lack of detail on data and statistics making it impossible for this reviewer to judge the validity of the conclusions.

This is regrettable since the study presents an admirable methodological tour-de-force combining electrophysiology, Ca⁺⁺-imaging and optogenetic manipulations during behaviour in vivo. Clearly, a lot of work has gone into the experiments. A rich, interconnected data set emerged that warrants in-depth, rigorous analysis. I wished the resulting data and its link to conclusions were presented with more care. Even more so as electrophysiology *and* imaging recordings from single units in CA3 during working memory processes are in short supply, underlining the potential broad interest of this study.

Overall, the manuscript is rather poorly written. The introduction omits essential context and relevant references. Methods are brief and do not provide the required insight into the analysis performed. The results section lacks critical experimental detail & statistics and includes interpretation & discussion of data. The discussion is woefully short lacking critical discussion of the data. But there is no point improving on this before establishing that the data supports the conclusions drawn. I will therefore focus below on major issues with the data, figure-by-figure, before summarising some additional general points:

Figure 1 / Supp Fig 1:

Using fiber-photometry from the left or right CA3 during a t-maze alternation task (t-DNMTP), this figure aims to provide evidence for: (1) CA3 activity increases in step with improving task performance; (2) a difference in activity between left and right CA3 when considering correct vs error trials in expert mice during the choice run.

I am puzzled by the Ca signal time courses during sample, delay and choice of the experiment. Fig1b does only add to this confusion. Why is there an increase in signal towards the end of the delay period? It would be helpful to see a plot fluorescent data from the entire experimental time course for all animals.

The baselines between correct and error trials (colour code legend is missing) in Fig1c/d are different. Panel 3 in in Fig1c/d is used to show a difference between left and right CA3. If the baselines were matched, would there still be the difference?

Supp Fig 1 suggests that there may be a systematic change in SNR of the fluorescent signal over the course of the experiment. Could this be an issue with the method? Photodamage or bleaching? Could the authors provide a panel like Supp Fig 1d for the eGFP control animals and quantitatively compare changes in SNR over the course of the experiment between the two groups?

Figure 2:

Using tetrode recordings, this figure aims to provide evidence for a difference between left or right CA3 in (1) the number of responsive neurons and (2) the relative timing of their peak activity.

The methods lack sufficient detail to fully understand how the data was analysed and z-scores generated. A lack of responsive cells may easily originate with recording / analysis technique. Was the

analysis carried out blind? Fig 2b/f suggest qualitative differences in the left/right CA3 data. I also remain unconvinced by the alleged difference in peak z-score timing. The stated p value is high. It is unclear whether the data was pooled from different sessions in the same or different animals. A repeated measures ANOVA followed by post-hoc tests may be more appropriate to explore these data.

Figure 3 / Supp Fig 3:

Using a combination of optogenetic suppression and contralateral imaging of CA3 activity, this figure aims to provide evidence for a difference between left and right CA3 during the choice element of the t-DNMTP task.

Data from Fig3b-c looks robust but again lacks appropriate statistics. The legends mentions a mix of paired and unpaired t-tests? RM ANOVA should be used.

As for Fig1. I do not follow the baseline normalisation/alignment used for the imaging data in Fig3e-f. The reported effects critically depend on this alignment and, as is, remain unconvincing.

These figures also present data from (1) PV+ neuron activation during the t-DNMTP task and (2) contralateral CA3 excitation. While interesting, the rationale for these experiments is very poorly explained in the manuscript. Both experiments come with a large number of confounds given the rich repertoire of PV IN action in the hippocampus and the densely recurrent architecture of CA3. Appropriate controls and discussion is required.

Figure 4:

Using monosynaptic retrograde rabies tracing, this figure aims to provide evidence for PV+ neurons in the medial septum providing more unilateral input to left or right CA3 than cholinergic medial septum neurons. As before, the methods lack details, manuscript and figures lack required data and statistics to fully evaluate this finding. 3 repeats per condition seem low, too. Confounds with rabies tracing would warrant more discussion.

Figure 5:

Using ontogenetic manipulation, this figure aims to provide evidence for PV+ neurons in the medial septum being an up-stream source of lateralised function between left and right CA3.

Activation of PV+ neurons in the medial septum using 20 Hz light flashes (duration?) seems to have no effect on the activity in left vs right CA3 (Fig5b). A switch to 80 Hz light flashes (why?) also results in no apparent difference (Fig5c-d). Suppression of PV+ neurons in the medial septum synapsing with left but not right CA3 neurons during the choice phase of the t-DNMTP task impairs performance yet no data or stats are provided in legend/manuscript to evaluate this claim.

General points:

- The manuscript makes abundant statement of significance or suggests trends (e.g. "correlated tightly"; "significant difference") without providing data or statistics;
- It is unclear throughout the manuscript whether data and comparisons are done repeats per animal or experiments from different animals. Please state clearly the data sources / origin of n;
- All figures will benefit from titles;
- Scale bars missing;
- Axis labels unclear / missing;
- Paired and unpaired t-tests are used inappropriately. Most data would better be analysed with repeated measures ANOVA followed by post-hoc tests, as appropriate;
- Methods lacking critical detail;
- Light stimulation intensity ranges from 1-60 mW often delivered continuously for 5 seconds: how was this measured and calibrated? what are the potential confounds of prolonged light stimulation / the limitation of the manipulation to specific time points?

Reviewer #2 (Remarks to the Author):

Review on "The lateralization of left hippocampal CA3 during the retrieval of spatial working memory" by Song et al for Nat. Comm. (2019)

Description of the study:

In this study, Song and col. attempt to unveil the contribution of left and right CA3 neural activity and cell properties, afferent circuits and cell types that support spatial working memory (SWM) tasks. Through a massive number of experiments using behavioral, functional, tracing and optogenetic approaches, the authors conclude that GABAergic drive from medial septum to the left hippocampus modulates neural activity associated with a correct prediction to solve a SWM task.

General Comments:

This reviewer is greatly impressed by the experiments' designs and extent of experimental work produced by the authors. I would endorse its publication pending the addressing of specific comments (below) and, crucially, extensive proofreading –unfortunately, grammatical mishaps did a major disservice to the manuscript.

Specific Comments:

Figure 1: My main questions here are: How could authors confidently report differences occur at the choice phase given baseline shifts extend from the delay phase? Are peak amplitude comparisons absolute (with baseline correction), or can the authors provide a rationale for their current comparisons?

Fig 1b: Could the authors provide an estimate of how many neurons are being recorded, and, particularly, describe whether signal surges come from increased activity within the same cells, more cells active or both?

Fig 1c: Could the authors indicate at what day of training these data were taken? I would assume after 5 days (>80% task efficiency). This brings up another question, do changes in correct/error rates change over training time? Once again, as indicated above, baseline differences across choice and sample phases may account for the peak amplitude differences.

Fig 1g: Would it be informative to carry out a Pearson (R) correlation to quantify the CA3 neuron activity to accrued experience correlate?

Supp. Fig 1a and c: Were free-running mice recorded with a barrier (for 10s) at the start point of the main arm, as experienced by trained mice during the delay phase? If not, could the authors explain why this was not necessary?

In line 141: Authors should indicate that 71% corresponds to the percentage of responsive cells. This is only realized after reading the whole sentence and creates confusion (same for the description of Right CA3 percentages).

Figure 2: Authors might consider making Left/Right labels more conspicuous in the figure. At initial readings it was difficult to follow which CA3 side data was referring about.

Fig: 2a: Authors should indicate that these data are from unit recordings.

Fig. 2b and f: Authors delved deeper on the properties of CA3 cells with increased activity (e.g. place, prediction). But, could authors also elaborate on the potential relevance of the increase/decrease/non responsive cells fraction populations? Increase/decrease ratio is greater in CA3 right than CA3 left, which has more non responsive and decreased cells.

Fig. 2i: Is the time of firing peak for recorded neurons in Left CA3 earlier than that in Right CA3 for all task phases?

Line 164: It says, "asymmetric effects"; should it say "asymmetric distribution"?

Figure 3: Could authors report the efficacy of their opto-inhibition on cell activity? On this matter, I

wonder, could authors tease out an estimate of cell activity contributions (predictions vs. place) from their opto-inhibition experiments? Similarly, but perhaps to a lesser extent, opto-inhibition at the delay phase does impact early (~2s) prediction cell activity (also see comment for Supp. Fig. 3c). In addition, Could the authors comment on the magnitude of performance drop with opto-inhibition? It is below 80%, still a good performance, I would say. Is this expected?

Fig 3e and f: It seems that opto-inhibition in Left CA3 area does impact calcium signal in CA3 right. In addition, I would suggest to relocate figure labels for "ON/OFF" and "correct/error" to the plots on Fig. 3e.

Fig. 3g: Could the authors explain why they preferred the wDMTP task instead of doing a DMTP task in the T maze?

Fig. 3j: The time lag dependency for the toDNMTL task (2 vs 10s) could indicate Left CA3 contribution might be a time lag function. Would the authors consider speculating on this (i.e. Left CA3 role dependency on time lag)?

Supp. Fig 3a: The drop in performance is far superior when activating rather than inhibiting CA3 areas. Would the authors consider further commenting on these findings, especially, in terms of the lateralization of CA3 activities for SWM?

Supp. Fig 3c: The data plots with opto-inhibition for the last 5s of the delay phase seem to be missing.

Supp. Fig 3e: The figure legend does not describe what is on the right plot. Also, the authors might consider speculating on the impact of PV neurons activation on signal-to-noise ratio as well as excitatory/inhibitory balance in terms of neural coding in LCA3, but not in RCA3. Once could speculate that if coding relies on signal-to-noise ratio, PV neural activity opto-inhibition should maximize SWM task performance.

Supp. Fig 3g: The authors show that calcium signal in Right CA3 increased sharply along with the activation of Left CA3 pyramidal neurons. Was the inverse experiment done (activation of Right CA3 and calcium signal recording in Left LCA3)?

Fig4a: I believe the orange coloring should be restricted to the MS cell bodies, with red and green incoming axons. Also, please indicate which side is Left and which side is Right.

Fig. 4b,c, and f: Could authors estimate the proportion of input cells to each CA3 area?

Fig. 4f: Could the authors indicate what is the 30% remaining from the CamKIIa-Cre Anti-Chat analysis? The plot shows zero for anti-PV, and I assume (perhaps in error) this plot should show all co-localized CamKIIa-Cre cells and, hence, anti-Chat and anti-PV to add up to 100%.

Fig 4g: I wonder whether authors could have used a combined RV (synaptic) and retrovirus (all terminals) approach to better assess what is the source of the cholinergic drive. Perhaps, it is unfeasible?

Line 276: It says, "these data approved"; should it say, "these data demonstrated"?

Fig 5a: Authors should indicate in legend that it refers to PV-Cre animals.

Fig 5c: Could the authors clarify why they used 20Hz for home cage activation (Supp. Fig. 3g) and 80Hz (hyper-stimulation) for these studies?

Fig 5g: Could the authors identify which population of cells (i.e. increase, decrease or place) is/are affected with this manipulation? Also, do authors possess similar analyses for the delay and arm phases?

Line 329: Discussion, it says, "SWM are remain unknown"; should it say, "SWM are unknown" or "SWM remain unknown"?

Methods:

Photometry: Could authors estimate the area of tissue from which the signal is being recorded, and confirm that such surface lies within (and not beyond) area CA3?

Line 390: It says, "microsyringe at a certain speed"; should it say, "microsyringe at a constant speed"?

Line 438: It says, "50 mm"; should it say, "50 micrometers"?

Line 439 and 445: There are clerical typos for -20oC and 4oC, respectively.

Line 479, 482: It says "firing rate was significant increased"; should it say "firing rate was significantly increased"?

Single unit analysis: Could the authors indicate the percentage of cells that fit the overlap criteria between prediction and main arm firing neurons and distinguish them from "delay" prediction neurons? Also, could these neurons be another type of task-relevant cells? In addition, were decreased neurons found only in the main arm, or also before (e.g. late delay phase)?

T-maze for delayed no-match-to-place (tDNMTP) task: Could the authors indicate how the mouse came back to the start position? Did it do it by itself or was it carried by the experimenter?

Line 516: It says, "the mouse visited non-sample arm"; should it say "the mouse visited the non-sample arm"?

Reviewer #3 (Remarks to the Author):

In this manuscript, Song et al provide an extensive study of lateralisation of mouse CA3 function in spatial working memory (SWM). While lateralisation of hippocampal function has been widely investigated in humans and non-human primates, rodent lateralisation is relatively understudied. The authors find both correlational and causal evidence for a dominant role of LCA3 neurons, compared to RCA3 neurons, in the choice phase of a delayed-nonmatching-to-place (DNMTP) task, using calcium imaging and optogenetic inhibition/stimulation respectively. Moreover, they show a selective requirement for medial septum(MS) PV+ neural projections to the LCA3 (but not RCA3) for performance in the same task. Overall the study is mostly well-controlled and uses converging approaches to support a left dominance in SWM.

Major comments

1-The main factor not controlled for in this study is animal speed, when identifying CA3 correlates of SWM performance in figures 1 and 2, the authors need to investigate whether, and to what extent, the neurons predictive of choice do so because of differences in animal speed in correct compared to incorrect trials.

2-It is not clear how MS-PV neuron input (figures 4 and 5) relates to the observed lateralisation. Silencing an input to the left CA3 may disrupt activity there non-selectively, hence causing the behavioural impairment seen, or may alternatively prevent a SWM-selective process. In other words its not clear whether MS-PV input to the LCA3 is permissive or instructive for LCA3 function in SWM. These possibilities could be disambiguated by investigating whether silencing MS-PV input to LCA3 selectively prevents predictive activity during the choice phase of the DNMTP task or has a more global effect on activity of all neurons during the same phase. The former would suggest a specific instructive role in working memory, while the latter would imply a non-specific permissive role. Knowing which is true is critical for any conclusions about a role for MS inputs in lateralisation of CA3 function.

Minor comments

1-When comparing left and right CA3 effects, its important that the authors use an ANOVA and show whether there is an interaction between hemisphere and trial-type (e.g. in figure 1 c and d) or between hemisphere and manipulation (e.g. figure 3 b and c).

2-Stimulating LCA3 neurons seems to induce an irreversible effect on animal performance in the DNMTP task (supplementary figure 3e). The authors should explain this.

3-The statement "unilateral CA3 neurons are mainly projected from MS PV neurons, rather than MS cholinergic neurons," is not supported by the data which instead shows that most bilaterally projecting MS neurons are cholinergic (figures 4f,i,j). This distinction has to be made clear.

4-The text is hard to follow at times as figure subpanels are referenced out of order (e.g.

supplementary figure 3c referenced in line 174 while supplementary figures 3 a and b are referenced later in line 187), this should be addressed to allow better readability.

5- The words "no impact" should be replaced by "little impact" in line 211 since there is a slight effect on contralateral CA3 activity when silencing unilaterally with NphR as reported in figure 3e and f.

6-The authors should explain why 20Hz (beta), rather than 10 Hz (theta) or 30 Hz upwards (gamma) stimulation of LCA3 neurons was used in this study.

Reviewer: Mohamady El-Gaby

Dear reviewers:

We highly appreciate for the valuable comments from reviewers. We have tried our best to revise the manuscript entitled "The lateralization of left hippocampal CA3 during the retrieval of spatial working memory" (reference number: NCOMMS-19-31173-T). Those comments are very helpful for revising and improving our manuscript and they also provide important guidance for our research. Regarding the reviewer's comments, we have studied these comments very carefully, have responded to the reviewers' questions one by one, and have made corrections which we hope to satisfy the reviewers and meet with their approval. In general, we have rewritten the introduction part and added more relevant references; we also detailed methods, especially for the electrophysiology, and applied more appropriate analysis tools, including AVONA, RM-AVONA and post-Hoc tests, to reanalyze our data; meanwhile, we also revised the results and tried our best to provide a more appropriate conclusion; the discussion was also revised by further discussing the relationship among our work with other publications. In details, we give responses to the comments from three reviewers in details. Please check all the corrections in the revised manuscript and the responses to the reviewers' comments are as follows (the replies are highlighted in red).

The detailed responses to referees' comments are listed below.

Responses to comments from reviewer #1 in details:

Reviewer #1 (Remarks to the Author):

In this study, the authors set out to address two questions concerning the activity of hippocampus CA3 during working memory tasks: (1) do bilateral CA3 neurons have different firing patterns in the working memory process, and (2) is any potential difference between left and right CA3 the result of upstream projections from the Medial Septum. My major issue with the study stems from the lack of detail on data and statistics making it impossible for this reviewer to judge the validity of the conclusions.

This is regrettable since the study presents an admirable methodological tour-de-force combining electrophysiology, Ca²⁺-imaging and optogenetic manipulations during behavior *in vivo*. Clearly, a lot of work has gone into the experiments. A rich, interconnected data set emerged that warrants in-depth, rigorous analysis. I wished the resulting data and its link to conclusions were presented with more care. Even more so as electrophysiology *and* imaging recordings from single units in CA3 during working memory processes are in short supply, underlining the potential broad interest of this study. Overall, the manuscript is rather poorly written. The introduction omits essential context and relevant references. Methods are brief and do not provide the required insight into the analysis performed. The results section lacks critical experimental detail & statistics and

includes interpretation & discussion of data. The discussion is woefully short lacking critical discussion of the data. But there is no point improving on this before establishing that the data supports the conclusions drawn. I will therefore focus below on major issues with the data, figure-by-figure, before summarising some additional general points:

1) Figure 1 / Supp Fig 1: Using fiber-photometry from the left or right CA3 during a t-maze alternation task (t-DNMTP), this figure aims to provide evidence for: (1) CA3 activity increases in step with improving task performance; (2) a difference in activity between left and right CA3 when considering correct vs error trials in expert mice during the choice run. I am puzzled by the Ca signal time courses during sample, delay and choice of the experiment. Fig1b does only add to this confusion. Why is there an increase in signal towards the end of the delay period? It would be helpful to see a plot fluorescent data from the entire experimental time course for all animals.

Responses: Thank you very much for this valuable comment. We felt so sorry for the mistakes which should be avoided, and really thank the reviewer to point it out. Firstly, it is noted that this example figure was plotted by randomly selected data. We apologize that in this case, due to the experiments were done by hands rather than automatically, the time points of certain trials were not precisely recorded. That is to say, the duration for the delay phase was less than 10s and the door was open to allow mice entering the choice phase roughly at 9s. Thus, the increased signal for the sample data at the end of the delay phase was actually counted from 0s to 9s rather from 0s to 10s by mistakes. To solve this problem, we have re-analyzed the data and replaced the Fig. 1b with a more precise one (please refer to the new Fig1b in the revised manuscript). Meanwhile, all the data for all animals were attached for your reference (please see Response-Figure 1).

Secondly, since all the mice in this experiment were well-trained, especially at the late end of the delay phase, mice were prepared to get ready for running and making choices, so the slightly increased signal would also be explained by the single-unite recording data that a group of neurons were considered as predictable neurons, suggesting they were firing with increased activity before the end of delay phase.

Response-Figure 1: Data for Ca^{2+} signals in CA3 region of all tested animals at the sample, delay and choice phases (n=14, LCA3:1,3,6,7,9,10,13,14; RCA3:2,4,5,8,11,12).

2) The baselines between correct and error trials (colour code legend is missing) in Fig1c/d are different. Panel 3 in Fig1c/d is used to show a difference between left and right CA3. If the baselines were matched, would there still be the difference?

Responses: Thanks for this valuable comment. Regarding this issue, we would like to explain this in two parts: on the one hand, it is a fact of the objectiveness of baseline inconsistencies, which is not caused by data analysis. Since we used the same method for analyzing data from both left and right CA3, and there are non-subjective preferences during data analysis. On the other hand, we speculated that during the trials with correct

choices, LCA3 contains certain numbers of prediction neurons with increased activities before entering the choice phase. As reviewer's request, we revised and normalized the data against baseline, even so, the left-right differences still existed. Please refer to the modified Fig1C (the right one) in the revised manuscript.

3) Supp Fig 1 suggests that there may be a systematic change in SNR of the fluorescent signal over the course of the experiment. Could this be an issue with the method? Photodamage or bleaching? Could the authors provide a panel like Supp Fig 1d for the eGFP control animals and quantitatively compare changes in SNR over the course of the experiment between the two groups?

Responses: Thanks very much for this comment and this issue pointed out by the reviewer is very important. Regarding the reviewer's concern, the SNRs of the fluorescent signal from the time course of 1- 8days were re-analyzed and calculated following the formula as shown in the Response-Figure 2, the SNR for day 1 was weak due to low signal at the beginning of experiment; the SNR for day 2-8 were basically stable, so we considered the SNR does not affect the system. Photodamage is usually caused by the concentration of virus, but in our experiments, we diluted the virus with a titer of 2×10^{12} , which could avoid the effect of photodamage. Furthermore, we provided the data in Suppl Fig1e that shows the data of GFP control animals.

$$\text{Signal Noise Ratio (SNR)} = 10 \times \text{Log}_{10} \frac{\text{Calcium signal}}{\text{GFP signal}}$$

Response-Figure 2: The Signal-to-noise ratio (SNR) for the mice that did T maze task during day 1 to day 8.

4) Figure 2: Using tetrode recordings, this figure aims to provide evidence for a difference between left or right CA3 in (1) the number of responsive neurons and (2) the relative timing of their peak activity. The methods lack sufficient detail to fully understand how the data was analysed and z-scores generated. A lack of responsive cells may easily originate with recording / analysis technique. Was the analysis carried out blind?

Responses: thanks so much for these precious comments which will significantly improve the quality of our manuscript. Indeed, the analysis was carried out blind. Following the reviewer's suggestion, we optimized the statistical analysis of the data in Figure 2 and the methods with detailed illustrations and the generation of z-scores have been added (please refer to the methods on line 534-557 in the revised manuscript)(also refer to the literature: Bolkan SS, 2017).

Reference:

Bolkan SS, Stujenske JM...Kellendonk C, Thalamic projections sustain prefrontal activity during working memory maintenance. *Nat Neurosci.* 2017 Jul;20(7):987-996. doi:10.1038/nn.4568.

5) Fig 2b/f suggest qualitative differences in the left/right CA3 data. I also remain unconvinced by the alleged difference in peak z-score timing. The stated p value is high. It is unclear whether the data was pooled from different sessions in the same or different animals. A repeated measures ANOVA followed by post-hoc tests may be more appropriate to explore these data.

Responses: Thank the review for pointing out these issues which are quite important for improving our manuscript. During our experiment, we have done the single-unit recordings on different animals. Regarding the p-value, it was originally calculated from comparing the time of firing peak between LCA3 and RCA3, with careful consideration of the reviewer's comment, we think comparison of the time of firing initiation in LCA3 and RCA3 would be more appropriate than the time of firing peak. Therefore, we re-analyzed the data by calculating the time of firing initiation which is more convincing and the p-value obtained is less than 0.0001 (Mann Whitney U test, ****p<0001) (see the line 177-178, 806-811 of the revised manuscript).

It is a very professional suggestion that we should use rm-ANOVA test in our data. We have revised all behavioral tests to apply rm-ANOVA followed by post-hoc tests. Please refer to our figures and the details in figure legends in the revised manuscript (line 823-920; 991-1002).

6) Figure 3 / Supp Fig 3: Using a combination of optogenetic suppression and contralateral imaging of CA3 activity, this figure aims to provide evidence for a difference between left and right CA3 during the choice element of the t-DNMTP task. Data from

Fig3b-c looks robust but again lacks appropriate statistics. The legends mentions a mix of paired and unpaired t-tests? RM ANOVA should be used.

Responses: Thanks very much for this comment and we agree with the reviewer. As suggested by the reviewer, the RM-ANOVA has been used for the data analysis and the details have been added in the legends of Figure 3 in the revised manuscript (see line 823-880).

7) As for Fig1. I do not follow the baseline normalisation/alignment used for the imaging data in Fig3e-f. The reported effects critically depend on this alignment and, as is, remain unconvincing.

Responses: Thanks so much for the comment. This issue is very important for improving the quality of our manuscript. Regarding this issue, we would like to explain this in two parts: on the one hand, it is a fact of the objectiveness of baseline inconsistencies, which is not caused by data analysis. Since we used the same method for analyzing data from both left and right CA3, and there are non-subjective preferences during data analysis. On the other hand, we followed the reviewer's suggestion and re-analyzed the data by normalization against baseline, and the results showed on the right-side of the revised Fig.3e-f, the left-right differences kept unchanged. Please refer to Fig3e-f (the right one) in the revised manuscript (see line761-767).

8) These figures also present data from (1) PV+ neuron activation during the t-DNMTP task and (2) contralateral CA3 excitation. While interesting, the rationale for these experiments is very poorly explained in the manuscript. Both experiments come with a large number of confounds given the rich repertoire of PV IN action in the hippocampus and the densely recurrent architecture of CA3. Appropriate controls and discussion is required.

Responses: Thanks very much for this valuable comment. We felt so sorry for the confounds caused in this manuscript.

The neuronal excitability, firing and synaptic integration of CA3 pyramidal neurons could be manipulated by receiving strong and diverse GABAergic inputs. It is reported some types of inhibitory neurons, such as PV-positive cells, PV-containing axo-axonic cells, could control the spiking of CA3 pyramidal neurons (Gulyás AI 2013, Rebola N 2017). It is known that CA3 contains many types of Interneurons, which have different functions on regulating CA3 pyramidal neurons. Activation of PV-positive INs was found to be highly effective in suppressing pyramidal neuronal activity, while VIP-positive neuron activation could both inhibit and disinhibit pyramidal neurons, the latter of which was realized by inhibiting SST- and PV-positive INs (Pi, Hangya et al. 2013, Zhang, Xu et al. 2014). This is a strong support for our selection of activating PV interneurons as a choice for

suppressing pyramidal neurons in LCA3. As indicated, the SWM was affected by activation of PV interneurons similarly as that only by inhibition of LCA3 pyramidal cells.

CA3 circuits form extensive excitatory interconnections between CA3 pyramidal neurons through associational and commissural (A/C) fiber synapses. This recurrent circuit is also proposed to be crucial for associative memory storage and retrieval (Rebola N, 2017). Our data suggested inhibition of LCA3 pyramidal neurons, rather than RCA3, could reduce the success rates of mice in the choice phase of tDNMTP tasks. However, activation of either LCA3 or CA3 pyramidal neurons in the choice phase impairs the task performances. We speculated that the activation of RCA3 pyramidal neurons might interfere the intrinsic neural activity of LCA3 and this was confirmed by the results that the GCamp6s signals in LCA3 increased sharply along with the activation of RCA3 pyramidal neurons. Meanwhile, we also performed experiments to exclude the effects of contralateral inhibition of CA3 neurons on ipsilateral CA3 neural activity. The data suggested that suppression of the ipsilateral CA3 neuronal activity in the choice phase seldom affects the contralateral CA3 neuronal activity. These data strongly implicated the critical role of LCA3 neurons, but not RCA3 neurons, on the retrieval of SWM.

This part has been added in the discussion in the revised manuscript (see line 357-397).

References:

Hájos N1, Karlócai MR, Gulyás AI. etc., Input-output features of anatomically identified CA3 neurons during hippocampal sharp wave/ripple oscillation in vitro. *J Neurosci.* 2013 Jul 10;33(28):11677-91. doi: 10.1523/JNEUROSCI.5729-12.2013.

Rebola N, Carta M, Mulle C., Operation and plasticity of hippocampal CA3 circuits: implications for memory encoding., *Nat Rev Neurosci.*, 2017 Apr; 18(4):208-220. doi:10.1038/nrn.2017.10.

Pi, H.J., et al., Cortical interneurons that specialize in disinhibitory control. *Nature*, 2013. 503(7477): p. 521-4.

Zhang, S., et al., Selective attention. Long-range and local circuits for top-down modulation of visual cortex processing. *Science*, 2014. 345(6197): p. 660-5.

9) Figure 4: Using monosynaptic retrograde rabies tracing, this figure aims to provide evidence for PV+ neurons in the medial septum providing more unilateral input to left or right CA3 than cholinergic medial septum neurons. As before, the methods lack details, manuscript and figures lack required data and statistics to fully evaluate this finding. 3 repeats per condition seem low, too. Confounds with rabies tracing would warrant more discussion.

Responses: We appreciate a lot for this reviewer pointing out these comments which will help us improve the quality of our manuscript. As suggested, we have updated the methods with details and used more proper statistical analysis to illustrate our main finding in Figure 4. In addition, we have increased the numbers of sampling from 3 repeats per

condition to 9 repeats for data analysis. In addition, proper discussion following the results has been added on line 272-284 of the revised manuscript.

10) Figure 5: Using ontogenetic manipulation, this figure aims to provide evidence for PV+ neurons in the medial septum being an up-stream source of lateralised function between left and right CA3. Activation of PV+ neurons in the medial septum using 20 Hz light flashes (duration?) seems to have no effect on the activity in left vs right CA3 (Fig5b). A switch to 80 Hz light flashes (why?) also results in no apparent difference (Fig5c-d). Suppression of PV+ neurons in the medial septum synapsing with left but not right CA3 neurons during the choice phase of the t-DNMTP task impairs performance yet no data or stats are provided in legend/manuscript to evaluate this claim.

Responses: Many thanks to the reviewer. We really agree with the reviewer and these issues pointed out by the reviewer are very important. In this experiment, 20Hz stimulation could only activate very few numbers of neurons while 80Hz stimulation could activate more neurons which is much easier for analyzing the neuronal activities. In addition, due to this experiment was done in homecage that mice perform non task-relevant behavior, so the neuronal activities had no apparent differences between LCA3 and RCA3 either by 20Hz stimulation or by 80Hz stimulation. This is a reflection of our finding that the MS^{PV} -LCA3 pathway is associated with task-dependent behaviors. However, in order to avoid misunderstandings, the original Figure 5b-c that were plotted from 80Hz stimulation has been replaced by the figure plotted from data of 20Hz stimulation, which has also been moved to the supplementary Figure 4b-c. please refer to the line 304-313; 967-984 of the revised manuscript.

In addition, we added new figure plotted from the data of T-maze behavioral tests that when MS^{chat} -CA3 pathways were manipulated. The data showed that inhibition of neither MS^{chat} -LCA3 nor MS^{chat} -RCA3 pathways have any impact on the retrieval of the SWM. This data has been added into Figure 5d-f in the revised manuscript.

11) General points:-The manuscript makes abundant statement of significance or suggests trends (e.g. “correlated tightly”; “significant difference”) without providing data or statistics;

Responses: Thanks for these valuable comments. These issues are very important and we have tried our best to amend these inappropriate words.

12) - It is unclear throughout the manuscript whether data and comparisons are done repeats per animal or experiments from different animals. Please state clearly the data sources / origin of n;

Responses: thanks for the comments. We have added relevant information about the data sources and the numbers of mice we used for experiments. Please refer to each figure legend in the revised manuscript.

13) - All figures will benefit from titles;

Responses: Thanks for the comments. Figure titles have been added in front of each figure legend. Please refer to the revised manuscript.

14) - Scale bars missing; - Axis labels unclear / missing;

Responses: thanks for the comments. Scale bars or unclear axis labels were all revised.

15) - Paired and unpaired t-tests are used inappropriately. Most data would better be analysed with repeated measures ANOVA followed by post-hoc tests, as appropriate;

Responses: thanks for the comments. We have re-analyzed the data using appropriate analysis tool, such as RM-ANOVA by post-hoc tests and Wilcoxon's signed-rank test as well as Mann-Whitney-U-test for nonparametric data. The related statistical details have been added in each figure legend. Please refer to the revised manuscript of line 630-636).

16) - Methods lacking critical detail

Responses: Thanks for the comments. Methods have been modified, especially we added the details of single-unit recording, MS^{PV/Chat}-CA3 recording and the statistical analysis, please refer to the line 534-557 and 630-636 in the revised manuscript.

17) - Light stimulation intensity ranges from 1-60 mW often delivered continuously for 5 seconds: how was this measured and calibrated? what are the potential confounds of prolonged light stimulation / the limitation of the manipulation to specific time points?

Responses: Thanks the reviewer for pointing this out. This is a very good question. We delivered the light when the door was open during the sample or the choice phases, for well-trained mice, once the door was open, the mice run immediately into the main arm, and the time durations in the choice phase were generally covered within the time of light stimulation. Please refer to the data added in the Response-Figure 3 in the following.

Response-Figure 3: Statistical analysis of the time duration of mice from initiating to run to getting the reward in choice phase.

Responses to comments from reviewer #2 in details:

Reviewer #2 (Remarks to the Author):

Review on “The lateralization of left hippocampal CA3 during the retrieval of spatial working memory” by Song et al for Nat. Comm. (2019)

Description of the study:

In this study, Song and col. attempt to unveil the contribution of left and right CA3 neural activity and cell properties, afferent circuits and cell types that support spatial working memory (SWM) tasks. Through a massive number of experiments using behavioral, functional, tracing and optogenetic approaches, the authors conclude that GABAergic drive from medial septum to the left hippocampus modulates neural activity associated with a correct prediction to solve a SWM task.

General Comments:

This reviewer is greatly impressed by the experiments' designs and extent of experimental work produced by the authors. I would endorse its publication pending the addressing of specific comments (below) and, crucially, extensive proofreading –unfortunately, grammatical mishaps did a major disservice to the manuscript.

Specific Comments:

18) Figure 1: My main questions here are: How could authors confidently report differences occur at the choice phase given baseline shifts extend from the delay phase? Are peak amplitude comparisons absolute (with baseline correction), or can the authors provide a rationale for their current comparisons?

Responses: Thank you for your instructive comments. We agree with the reviewer and these issue pointed out by the reviewer are very important. It is a fact of the objectiveness of baseline inconsistencies, which is not caused by data analysis. We used the same method for analyzing data from both left and right CA3 without any subjective preferences. We speculated that during the trials with correct choices, LCA3 contains certain numbers of prediction neurons with increased activities before entering the choice phase. Even though, when the data was normalized against baseline, the left-right differences still existed. Please refer to Fig1C (the right one) in the revised manuscript.

19) Fig 1b: Could the authors provide an estimate of how many neurons are being recorded, and, particularly, describe whether signal surges come from increased activity within the same cells, more cells active or both?

Responses: Thanks so much for this comment. According to the immunofluorescent staining, it can be shown that there are approximately $53.64 \pm 18.07\%$ neurons in CA3 (GFP⁺/DAPI⁺) that are recorded. However, it is noted that, due to the fact that Ca²⁺

imaging recording could not reflect cellular resolution, so it is hard to say the recorded neuronal activities are from the same cells or from more active cells. But, the data obtained by single-unit recording could tell that most neurons recorded (LCA3: 71.9%; RCA3: 66.9%) are overlapped neurons with increased activities during the sample and the choice phase, please refer to the Response-Figure 4 in the following.

Response-Figure 4: Statistical analysis of the overlapped neurons with increased activities during the sample and the choice phase.

20) Fig 1c: Could the authors indicate at what day of training these data were taken? I would assume after 5 days (>80% task efficiency). This brings up another question, do changes in correct/error rates change over training time? Once again, as indicated above, baseline differences across choice and sample phases may account for the peak amplitude differences.

Responses: Thanks so much for this comment. As assumed by the reviewer, these data were taken from day 6 to day 8. We calculated the error/correct rates during day 6 to day 8 and found that, the rates kept unchanged over the training time (see the Response-Figure 5). Regarding the issue of baseline inconsistencies, we would consider it an objective phenomenon which is not caused by data analysis. Even when we re-analyzed the data by normalization against baseline, the left-right differences still existed. Please refer to Fig1C (the right one) in the revised manuscript.

Response-Figure 5: Data for the error/correct rates of mice during day 6 to day 8

21) Fig 1g: Would it be informative to carry out a Pearson (R) correlation to quantify the CA3 neuron activity to accrued experience correlate?

Responses: Thanks very much for this comment. As suggested, we have optimized the statistical analysis by carrying out a Pearson correlation, and please refer to the line 774 in the revised figure 1 legend.

22) Supp. Fig 1a and c: Were free-running mice recorded with a barrier (for 10s) at the start point of the main arm, as experienced by trained mice during the delay phase? If not, could the authors explain why this was not necessary?

Responses: Thanks so much for this comment. Recording this question, we did this behavioral test and found that under non working-memory related task, the well-trained mice did not show any increased fluorescent signal at the time point of choice begin, suggesting the fluorescent signal is working-memory task-dependent (please refer to the Suppl Fig 1d). The relevant result has been added on line 144-146; 785-787 of the revised manuscript.

23) In line 141: Authors should indicate that 71% corresponds to the percentage of responsive cells. This is only realized after reading the whole sentence and creates confusion (same for the description of Right CA3 percentages).

Responses: Thanks for your kind suggestion. The part has been revised into "Overall, 70.9% of recorded cells were accounted for the responsive cells in LCA3 (62.56%, 142 of 227 units are increased; 8.37%, 19 of 227 units decreased), similarly, 70.7% were responsive cells in RCA3 (65.64%, 128 of 197 units are increased; 5.13%, 10 of 197

units decreased). There were no significances for non-responsive cells in LCA3 or RCA3.” Please refer to the line 158-162 in the revised manuscript.

24) Figure 2: Authors might consider making Left/Right labels more conspicuous in the figure. At initial readings it was difficult to follow which CA3 side data was referring about.

Responses: Thanks so much for this comment. We are so sorry for the unclear labels. We have revised the labels in figure 2, please refer to the updated figure 2 in the revised manuscript.

25) Fig: 2a: Authors should indicate that these data are from unit recordings.

Responses: Thanks so much for this comment. A proper title “single-unit recording of neurons in bilateral CA3 during the choice phase” has been added to legends of the Figure 2. And similarly, all the figures were given short titles in the revised manuscript (refer to the line of 750-751, 766-767, 793, 814, 822-823, 884-885, 941, 967-968, 986-987).

26) Fig. 2b and f: Authors delved deeper on the properties of CA3 cells with increased activity (e.g. place, prediction). But, could authors also elaborate on the potential relevance of the increase/decrease/non responsive cells fraction populations? Increase/decrease ratio is greater in CA3 right than CA3 left, which has more non responsive and decreased cells.

Responses: Thanks very much for this valuable comment. As suggested by the reviewer 1, we optimized the methods for statistical analysis and the proportions of increase/decrease/non-responsive cells were described in line 158-162 and 171-173, in the revised manuscript. Though asymmetrical distribution of these cells occurred in LCA3 and RCA3, the increase/decrease ratio seems similar in bilateral CA3.

27) Fig. 2i: Is the time of firing peak for recorded neurons in Left CA3 earlier than that in Right CA3 for all task phases?

Responses: Thanks the reviewer for pointing out this question, it is very interesting. We analyzed the data in the sample phase and found that, the time of firing initiation in LCA3 is earlier than that in RCA3, which is similar to the data in the choice phase (please refer to the Response-Figure 6 in the following). It is noted that, in order to more precisely illustrate our data, “the time of firing peak” has been revised into “the time of firing initiation”, please refer to the line 177-179 and 807-809 in the revised manuscript.

Response-Figure 6: the time of firing initiation of CA3 neurons in the sample phase

28) Line 164: It says, “asymmetric effects”; should it say “asymmetric distribution”?

Responses: Thanks for this comment. It has been revised as suggested.(line 186)

29) Figure 3: Could authors report the efficacy of their opto-inhibition on cell activity? On this matter, I wonder, could authors tease out an estimate of cell activity contributions (predictions vs. place) from their opto-inhibition experiments? Similarly, but perhaps to a lesser extent, opto-inhibition at the delay phase does impact early (~2s) prediction cell activity (also see comment for Supp. Fig. 3c).

Responses: Thanks so much for the comment. It is indeed a key point. We spent over two months on doing the experiments that the reviewer suggested. We tried to apply op-tetrode to inhibit MS^{PV} neurons and record the neuronal activities in bilateral CA3 during the t-

DNMTP task, unfortunately, we could not get robust signals due to some obstacles that the experiment itself hardly be avoided, such as the noises caused by free-moving mice and the twisted op-tetrode during the experiment. We felt so sorry for this. However, we are still working on it and hopefully make it possible in the near future.

30) In addition, Could the authors comment on the magnitude of performance drop with opto-inhibition? It is below 80%, still a good performance, I would say. Is this expected?

Responses: Thanks so much for this comment. Many studies have reported that multiple brain regions are functional to manipulate learning and memories (Bolkan SS, 2017; Spellman T, 2015), so does the working memory. Working memory is regulated by CA3 but not fully dependent on CA3. We suspected that even inhibition of CA3 region would impair a part of spatial information, therefore, it can be expected that inhibition of CA3 would reduce the ability of working memory in certain degree that mice still could show a good performance.

References:

Bolkan SS, Stujenske JM...Kellendonk C, Thalamic projections sustain prefrontal activity during working memory maintenance. *Nat Neurosci.* 2017 Jul;20(7):987-996. doi: 10.1038/nn.4568.

Spellman T, Rigotti M...Gordon JA, Hippocampal-prefrontal input supports spatial encoding in working memory. *Nature.* 2015 Jun 18;522(7556):309-14. doi: 10.1038/nature14445.

31) Fig 3e and f: It seems that opto-inhibition in Left CA3 area does impact calcium signal in CA3 right. In addition, I would suggest to relocate figure labels for "ON/OFF" and "correct/error" to the plots on Fig. 3e.

Responses: Thanks so much for pointing this out. Though inhibition of LCA3 caused slight decrease of Ca²⁺ signal in RCA3, the Ca²⁺ signals in correct and error trials showed no differences, which is consistent with light-off trials. In addition, we have relocated the figure labels to avoid misunderstanding. Please refer to the Figure 3 in the revised manuscript. (line 821)

32) Fig. 3g: Could the authors explain why they preferred the wDMTP task instead of doing a DMTP task in the T maze?

Responses: Thanks very much for this comment. In this study, we initially examined the lateralization of LCA3 in SWM by using T-Maze task; afterwards, in order to make sure this phenomenon is prevalent in mice, we applied both wDMTP and toDNMTL tasks. All these behavioral tests confirmed the common phenomena of the lateralization of LCA3 in SWM.

33) Fig. 3j: The time lag dependency for the toDNMTL task (2 vs 10s) could indicate Left CA3 contribution might be a time lag function. Would the authors consider speculating on this (i.e. Left CA3 role dependency on time lag)?

Responses: We really appreciate a lot for this comment. It is really an interesting point. Gordon J team applied 60 s delay T-maze working memory task to investigate the function of vHPC-mPFC and MD-mPFC neuronal circuits in working memory. In addition, Jung MW team used 3 s delay T-maze working memory task to study the role of PV- and SOM-interneurons in mPFC in spatial working memory. These work indicated that different brain regions show distinct sensitivity of delay time lag in working memory studies. In our experiments, under toDNMTL tasks, 2s and 5s delay showed different performance. Regarding this issue, we would very like to consider the contribution of CA3 on time lag function in our future work.

References:

Bolkan SS, Stujenske JM...Kellendonk C, Thalamic projections sustain prefrontal activity during working memory maintenance. *Nat Neurosci.*2017 Jul;20(7):987-996. doi:10.1038/nn.4568.

Dohoung Kim, Huijeong Jeong...Min WhanJung, Distinct Roles of Parvalbumin- and Somatostatin- Expressing Interneurons in Working Memory. *Neuron.* 2016 Nov 23; 92(4):902-915. doi: 10.1016/j.neuron.2016.09.023.

34) Supp. Fig 3a: The drop in performance is far superior when activating rather than inhibiting CA3 areas. Would the authors consider further commenting on these findings, especially, in terms of the lateralization of CA3 activities for SWM?

Responses: Thanks very much for this reminder. We have reviewed certain publications and further commented on this. The CA3 has been reported to play specific roles in hippocampal function and memory, especially for the sparse inputs from the dentate gyrus to CA3 and for the extended local recurrent connectivity that operates the CA3 autoassociative network and accordingly forms extensive excitatory interconnections between CA3 pyramidal cells through associational and commissural (A/C) fiber synapses (Rebola N, 2017). This recurrent circuit is also proposed to be crucial for associative memory storage and retrieval. In addition, the synapses from mossy fibers to CA3 pyramidal cells were considered to display a wide dynamic range of short-term plasticity. Considering these, it is undoubtedly that activation of CA3 pyramidal neurons would disturb the internal neural activity of CA3, including CA3 autoassociative networks and the interconnections that CA3 forms with other brain regions. This was confirmed by the results that the GCamp6s signals in LCA3 increased sharply along with the activation of RCA3 pyramidal neurons.

Reference:

Rebola N, Carta M, Mulle C. Operation and plasticity of hippocampal CA3 circuits: implications for memory encoding. *Nat Rev Neurosci.* 2017 Apr; 18(4):208-220. doi:10.1038/nrn.2017.10.

35) Supp. Fig 3c: The data plots with opto-inhibition for the last 5s of the delay phase seem to be missing.

Responses: Thanks very much for this comment. The data plots with opto-inhibition for the last 5s of the delay phase was displayed in result 3, please refer to the figure 3 legend in the revised manuscript (see line 830; 842).

36) Supp. Fig 3e: The figure legend does not describe what is on the right plot. Also, the authors might consider speculating on the impact of PV neurons activation on signal-to-noise ratio as well as excitatory/inhibitory balance in terms of neural coding in LCA3, but not in RCA3. Once could speculate that if coding relies on signal-to-noise ratio, PV neural activity opto-inhibition should maximize SWM task performance.

Responses: Thanks for this valuable comment. For supplementary Figure 3e, the right plot was added to the figure legend in the revised manuscript of line 929-933.

It is widely accepted that inhibitory neurons (mostly interneurons, INs) interact with each other and form neural circuitry of inhibitory synaptic connections. Meanwhile, these interneurons also reciprocally connect with and regulate other excitatory neurons (Donato, Rompani et al. 2013, Pfeffer, Xue et al. 2013, Letzkus, Wolff et al. 2015). Activation of PV-positive INs was found to be highly effective in suppressing pyramidal neuronal activity, while VIP-positive neuron activation could both inhibit and disinhibit pyramidal neurons, the latter of which was realized by inhibiting SST- and PV-positive INs (Pi, Hangya et al. 2013, Zhang, Xu et al. 2014).

It was also reported that, different subtypes of inhibitory interneurons function distinctively in different task-related performances (Pinto and Dan 2015, Kamigaki and Dan 2017). Dan and his team has revealed that, SOM-positive interneurons are responsible for signaled motor action, while VIP-positive interneurons highly respond to action outcomes, while PV-positive interneurons less selectively respond to either sensory cues, motor actions or trial outcomes (Pinto and Dan 2015). In 2017, Dan further claimed that suppression of pyramidal neuronal activity by activation of SOM- or PV-positive interneurons could impair the memory-guided task performance, while activation of VIP-positive interneurons function oppositely that could induce the task performance and neural coding of action plans (Kamigaki and Dan 2017). This suggested that activation of VIP-positive interneurons could disinhibit pyramidal neurons by inhibiting SOM- or PV-positive interneurons, which in turn enhance the behavioral performance. This was quite consistent with the discovery of disinhibitory neural circuitry that was considered to be uniquely

controlled by inhibitory neurons that specifically suppress the firing of other inhibitory neurons. This part has been added to the discussion in line 363-385 of the revised manuscript.

Given the studies discussed above, it is not hard to understand our data showing that activation of PV neurons could inhibit the neural activity of pyramidal neurons, thus enhancing task performance in SWM. It is noted that T-maze is a classical but very simple behavioral test of SWM that results in high successful rates, it would be more convincing if more complicated tasks were applied, such as, the multiple-armed T-maze task, nevertheless of which would definitely affects the behavioral observation during task. So that is also why we selected to use the classic T-maze task for this study.

References:

Pfeffer, C.K., et al., Inhibition of inhibition in visual cortex: the logic of connections between molecularly distinct interneurons. *Nat Neurosci*, 2013. 16(8): p. 1068-76.

Donato, F., S.B. Rompani, and P. Caroni, Parvalbumin-expressing basket-cell network plasticity induced by experience regulates adult learning. *Nature*, 2013. 504(7479): p. 272-6.

Letzkus, J.J., S.B. Wolff, and A. Luthi, Disinhibition, a Circuit Mechanism for Associative Learning and Memory. *Neuron*, 2015. 88(2): p. 264-76.

Pi, H.J., et al., Cortical interneurons that specialize in disinhibitory control. *Nature*, 2013. 503(7477): p. 521-4.

Zhang, S., et al., Selective attention. Long-range and local circuits for top-down modulation of visual cortex processing. *Science*, 2014. 345(6197): p. 660-5.

Kamigaki, T. and Y. Dan, Delay activity of specific prefrontal interneuron subtypes modulates memory-guided behavior. *Nat Neurosci*, 2017. 20(6): p. 854-863.

Pinto, L. and Y. Dan, Cell-Type-Specific Activity in Prefrontal Cortex during Goal-Directed Behavior. *Neuron*, 2015. 87(2): p. 437-50.

37) Supp. Fig 3g: The authors show that calcium signal in Right CA3 increased sharply along with the activation of Left CA3 pyramidal neurons. Was the inverse experiment done (activation of Right CA3 and calcium signal recording in Left LCA3)?

Responses: Thanks very much for this comment. We have added the experiments the reviewer suggested. As shown in the Response-Figure 7 in the following, activation of RCA3 similarly resulted in increased neuronal activity of LCA3.

Response-Figure 7: the Ca^{2+} signal of LCA3 when RCA3 was stimulated

38) Fig4a: I believe the orange coloring should be restricted to the MS cell bodies, with red and green incoming axons. Also, please indicate which side is Left and which side is Right.

Responses: Thanks very much for this comment. We have revised the pattern diagram. For RV viruses, mCherry and GFP-labelled RVs were counterbalanced by injection into either left or right CA3, which would avoid the differences of expression levels. But the colors could not reflect the exact side of labeled neurons. The diagram of Fig. 4a has been modified as suggested (line 940).

39) Fig. 4b,c, and f: Could authors estimate the proportion of input cells to each CA3 area?

Responses: Thanks so much for this comment. Since we injected AAV-DIO-BFP-TVA in bilateral CA3 which is blue colored, we did not further stain with DAPI (which is blue colored), it is hard to count all the cells in CA3 in this experiment. Basically, there were approximately 2.1 ± 1.1 input cells (GFP or mCherry) from the bilateral CA3 co-localized with TVA (blue) in PV-cre mice; while, 7.8 ± 2.1 input cells (GFP or mCherry) that could be observed to co-localize with TVA (blue) in CamKII-cre mice (please refer to the Response-Figure 8).

Response-Figure 8: The histochemical staining of input cells of bilateral CA3 in PV-Cre and CamkIIα-Cre by rabies tracing system

40) Fig. 4f: Could the authors indicate what is the 30% remaining from the CamKIIa-Cre Anti-Chat analysis? The plot shows zero for anti-PV, and I assume (perhaps in error) this plot should show all co-localized CamKIIa-Cre cells and, hence, anti-Chat and anti-PV to add up to 100%.

Responses: Thanks very much for this comment. As is known that, the MS region contains several types of neurons, including chat neurons, PV interneurons and pyramidal neurons. So we speculated that the remaining 30% might be other types of neurons. Nevertheless, it is noted that the efficiency of histochemical staining cannot reach to 100%, so it is hard to say whether the co-localized CamKIIα-Cre cells, together with anti-Chat and anti-PV could be added up to 100%.

41) Fig 4g: I wonder whether authors could have used a combined RV (synaptic) and retrovirus (all terminals) approach to better assess what is the source of the cholinergic drive. Perhaps, it is unfeasible?

Responses: Thanks very much for this valuable comments. In our current tracing system, for AAVretro tracing experiments, AAV9-Ef1 α -DIO-GFP and AAV9-Ef1 α -fDIO-tdTomato were applied; for rabies tracing experiments, RV-EnVA-EGFP and RV-EnVA-mCherry were applied. Theoretically, if RV tracing and AAVretro tracing were combined to use, it requires at least five channel-labeled protein markers, which cannot be reached by current techniques (at least by our current devices). So we felt sorry that it seems unfeasible in reality so far yet.

42) Line 276: It says, “these data approved”; should it say, “these data demonstrated”?

Responses: Thanks for this comment. We have revised it. Please refer to the revised manuscript.

43) Fig 5a: Authors should indicate in legend that it refers to PV-Cre animals.

Responses: Thanks for this comment. We have revised it. Please refer to the revised manuscript.

44) Fig 5c: Could the authors clarify why they used 20Hz for home cage activation (Supp. Fig. 3g) and 80Hz (hyper-stimulation) for these studies?

Responses: Many thanks to this comment. In this experiment, 20Hz stimulation could only activate very few numbers of neurons while 80Hz stimulation could activate more neurons which is more conducive for analyzing the neuronal activities. In addition, due to this experiment was done in homecage that mice perform non-task relevant behavior, so the neuronal activities had no apparent differences either by 20Hz stimulation or by 80Hz stimulation. This was quite consistent with our finding that the MS^{PV}-LCA3 pathway is associated with task-dependent behaviors. In order to avoid misunderstandings, the original Figure 5b-c that were plotted from 80Hz stimulation has been replaced by the figure plotted from data of 20Hz stimulation, which has also been moved to the supplementary Figure 4b-c. please refer to the revised manuscript.

In addition, we added new figure plotted from the data of T-maze behavioral tests that when MS^{chat}-CA3 pathways were manipulated. The data showed that inhibition of neither MS^{chat}-LCA3 nor MS^{chat}-RCA3 pathways have any impact on the retrieval of the SWM (refer to the line 304-329;986-1002 in the revised manuscript). This data has been added into Figure 5d-f in the revised manuscript.

45) Fig 5g: Could the authors identify which population of cells (i.e. increase, decrease or place) is/are affected with this manipulation? Also, do authors possess similar analyses for the delay and arm phases?

Responses: Thanks very much for this valuable comment. To answer this question, we spent over two months on doing the experiments, trying to apply op-tetrode to inhibit MS^{PV} neurons and record the neuronal activities in bilateral CA3 during the t-DNMTP task, unfortunately, we could not get robust signals due to some obstacles that the experiment itself hardly be avoided, such as the noises caused by free-moving mice and the twisted op-tetrode during the experiment. However, we will definitely consider it as our main future work and will try our best to get the answer once we get more advanced techniques.

46) Line 329: Discussion, it says, "SWM are remain unknown"; should it say, "SWM are unknown" or "SWM remain unknown"?

Responses: Thanks for this comment, it has been revised. Please refer to the Line350 in the revised manuscript.

Methods:

47) Photometry: Could authors estimate the area of tissue from which the signal is being recorded, and confirm that such surface lies within (and not beyond) area CA3?

Responses: Thanks very much for this comment. Actually, the position for virus injection (the same place for signal recording, which is at AP: -2.06 mm; ML: ± 2.35 mm; DV: -2.35 mm) was mentioned in the method, which is the pyramidal cell layer (line 455-456). In addition, please also refer to the Stereotactic injection diagram in Fig1a.

48) Line 390: It says, "microsyringe at a certain speed"; should it say, "microsyringe at a constant speed"?

Responses: Thanks for this comment, it has been revised. Please refer to the Line 450 in the revised manuscript.

49) Line 438: It says, "50 mm"; should it say, "50 micrometers"?

Responses: Thanks for this comment, it has been revised (line 495).

50) Line 439 and 445: There are clerical typos for -20oC and 4oC, respectively.

Responses: Thanks for this comment, it has been revised (line 497, 503).

51) Line 479, 482: It says "firing rate was significant increased"; should it say "firing rate was significantly increased"?

Responses: Thanks for this comment, it has been revised.

52) Single unit analysis: Could the authors indicate the percentage of cells that fit the overlap criteria between prediction and main arm firing neurons and distinguish them from “delay” prediction neurons? Also, could these neurons be another type of task-relevant cells? In addition, were decreased neurons found only in the main arm, or also before (e.g. late delay phase)?

Responses: Thanks so much for this comment. As suggested by the first reviewer 1, we optimised the analysis methods for our data. As revised method of categorization, prediction neurons and main arm neurons can be distinguished well. The time point of neuron initiating to fire was used to classify the prediction neuron and mains arm neurons. (please refer to method, line 534-557)

For decreased neurons, 15 out of 19 decreased neurons in LCA3 and 8 of 10 in RCA3 were found in the main arm.

53) T-maze for delayed no-match-to-place (tDNMTP) task: Could the authors indicate how the mouse came back to the start position? Did it do it by itself or was it carried by the experimenter?

Responses: Thanks for your comment. We are so sorry that this was not illustrated very clearly. This was demonstrated as” the mice are allowed to return to the start point by themselves, however, if the mice could not return to the start point by themselves 15s after feeding, they will be forced to return to the start point”. Please refer to the line 573-576 in the revised manuscript.

54) Line 516: It says, “the mouse visited non-sample arm”; should it say “the mouse visited the non-sample arm”?

Responses: Thanks for this comment, it has been revised.

Responses to comments from reviewer #3 in details:

Reviewer #3 (Remarks to the Author):

In this manuscript, Song et al provide an extensive study of lateralisation of mouse CA3 function in spatial working memory (SWM). While lateralisation of hippocampal function has been widely investigated in humans and non-human primates, rodent lateralisation is relatively understudied. The authors find both correlational and causal evidence for a dominant role of LCA3 neurons, compared to RCA3 neurons, in the choice phase of a delayed-nonmatching-to-place (DNMTP) task, using calcium imaging and optogenetic inhibition/stimulation respectively. Moreover, they show a selective requirement for medial septum(MS) PV+ neural projections to the LCA3 (but not RCA3) for performance in the same task. Overall the study is mostly well-controlled and uses converging approaches to support a left dominance in SWM.

Major comments

55) 1-The main factor not controlled for in this study is animal speed, when identifying CA3 correlates of SWM performance in figures 1 and 2, the authors need to investigate whether, and to what extent, the neurons predictive of choice do so because of differences in animal speed in correct compared to incorrect trials.

Responses: Thanks very much for this comment. We felt so sorry that, due to our experimental condition, we only recorded the time duration rather than speed in the choice phase. From our data, the mice in LCA3 and RCA3 groups showed no significant differences of the time duration in the process of either the correct or wrong trials (please refer to the Response-Figure 9). Some research has also indicated the low relationship between speed and changes of LFP in spatial working memory (please refer to the literature: Duvarci S, 2018; Sasaki T, 2018).

Meanwhile, in optogenetic manipulation of T-maze working memory tasks, the choice duration was prolonged with significance when LCA3 was inhibited or activated, or RCA3 activated, in the choice phase (please refer to the Response-Figure 10). Mice showed hesitation in T-junction with light stimulation, which was barely seen even in error trials without light manipulation. Together, in normal condition (without optogenetic manipulation), speed does not make differences between correct and incorrect trials. While, not only the optogenetic interference of CA3 affected the performance, but also influenced the time durations of choice making. As suggested by this very previous comment, we will definitely consider the speed factor during our future studies.

References:

Duvarci S, Simpson EH...Sigurdsson T. Impaired recruitment of dopamine neurons during working memory in mice with striatal D2 receptor overexpression. *Nat Commun.* 2018 Jul 19;9(1):2822. doi: 10.1038/s41467-018-05214-4.

Sasaki T, Piatti VC...Leutgeb JK. Dentate network activity is necessary for spatial working memory by supporting CA3 sharp-wave ripple generation and prospective firing of CA3 neurons. Nat Neurosci. 2018 Feb; 21(2):258-269. doi: 10.1038/s41593-017-0061-5.

Response-Figure 9: the statistical analysis of the choice duration in the process of either the correct or wrong trials in fiber photometry tasks.

Response-Figure 10: the statistical analysis of the choice duration in the process of either the correct or wrong trials in optogenetic manipulation tasks.

56) 2-It is not clear how MS-PV neuron input (figures 4 and 5) relates to the observed lateralisation. Silencing an input to the left CA3 may disrupt activity there non-selectively, hence causing the behavioural impairment seen, or may alternatively prevent a SWM-selective process. In other words, its not clear whether MS-PV input to the LCA3 is permissive or instructive for LCA3 function in SWM. These possibilities could be disambiguated by investigating whether silencing MS-PV input to LCA3 selectively prevents predictive activity during the choice phase of the DNMTTP task or has a more global effect on activity of all neurons during the same phase. The former would suggest a specific instructive role in working memory, while the latter would imply a non-specific permissive role. Knowing which is true is critical for any conclusions about a role for MS inputs in lateralisation of CA3 function.

Responses: Thanks very much for this critical comment. Regarding this issue, we spent over two months on doing the experiments that the reviewer suggested. We tried to apply op-tetrode to inhibit MS^{PV} neurons and record the neuronal activities in bilateral CA3 during the t-DNMTTP task, unfortunately, we could not get robust signals due to some obstacles that the experiment itself hardly be avoided, such as the noises caused by free-moving mice and the twisted op-tetrode during the experiment. We felt so sorry for this. However, we are still working on it and hopefully make it possible in the near future

57) Minor comments: 1-When comparing left and right CA3 effects, it's important that the authors use an ANOVA and show whether there is an interaction between hemisphere and trial-type (e.g. in figure 1 c and d) or between hemisphere and manipulation (e.g. figure 3 b and c).

Responses: Thanks very much for this comment. We have applied ANOVA for data analysis in all behavioral tests. Please refer to the updated figure legends in the revised manuscript (line 823-920; 991-1002).

58) 2-Stimulating LCA3 neurons seems to induce an irreversible effect on animal performance in the DNMTTP task (supplementary figure 3e). The authors should explain this.

Responses: Thanks very much for this valuable comment. In our experiment, high-frequency 80Hz was applied for activation of PV neurons in LCA3, which could consequently inhibit the neuronal activities of pyramidal neurons. However, it is noted that high-frequency stimulation might cause certain damage of PV neurons, this might explain the irreversible effects on mice performance caused in this experiment.

59) 3-The statement “unilateral CA3 neurons are mainly projected from MS PV neurons, rather than MS cholinergic neurons,” is not supported by the data which instead shows

that most bilaterally projecting MS neurons are cholinergic (figures 4f,i,j). This distinction has to be made clear.

Responses: Many thanks for this valuable comment. As suggested, we have optimized this paragraph and make it much easier to understand. Please refer to the line 272-283 in the revised manuscript.

60) 4-The text is hard to follow at times as figure subpanels are referenced out of order (e.g. supplementary figure 3c referenced in line 174 while supplementary figures 3 a and b are referenced later in line 187), this should be addressed to allow better readability.

Responses: Thanks very much for this comment. This will improve the quality of our manuscript. Regarding this, we have relabeled the subpanels in supplementary Fig.3 to improve the readability. Please refer to line 198-212 and 883 in the revised manuscript.

61) 5- The words “no impact” should be replaced by “little impact” in line 211 since there is a slight effect on contralateral CA3 activity when silencing unilaterally with NphR as reported in figure 3e and f.

Responses: Thanks for this comment. It has been revised as suggested. Please refer to the line 231 of the revised manuscript.

62) 6-The authors should explain why 20Hz (beta), rather than 10 Hz (theta) or 30 Hz upwards (gamma) stimulation of LCA3 neurons was used in this study.

Responses: Thanks for the precious comment. We had gone through few publications which talks about different bands of neuronal firing would be affected during mouse behavioral performances. It was reported that the β 2 oscillations (23-23Hz) in CA3 showed strong bursts when mice explore novel, rather than familiar, environment; and the theta (10Hz) and beta (20Hz) band synchronization of neuronal firing patterns and local field potential activity in mice CA1 increased in the reward cue tasks (Lansink CS, 2016; Berke JD, 2008). From our analysis of the local field potential activity, we found that mice at the sample and the choice phases showed significant neuronal firing changes at 20Hz (data not shown), that is the reason why we selected 20Hz for stimulation of MS neurons to observe their neuronal firing changes.

References:

Lansink CS, Meijer GT....Pennartz CM, Reward Expectancy Strengthens CA1 Theta and Beta Band Synchronization and Hippocampal-Ventral Striatal Coupling. J Neurosci. 2016 Oct 12;36(41):10598-10610.

Berke JD, Hetrick V...Greene RW. Transient 23-30 Hz oscillations in mouse hippocampus during exploration of novel environments. *Hippocampus*. 2008;18(5):519-29. doi: 10.1002/hipo.20435.

Finally, we again thank the reviewers for their hard work and careful consideration on our manuscript. Those valuable and precious comments will help us improve our work significantly. We hope our responses could well answer the reviewers' questions and meet the approval for publication. Thanks again!

Reviewers' comments:

Reviewer #1 (Remarks to the Author):

I thank the authors for their point-by-point response to my earlier comments. I shall reply in kind:

Point 1:

I appreciate the response figure which shows time courses for the entire dataset. This is very informative. I encourage the authors to include this either in Fig 1 or in a supplementary figure. Thank you also for clarifying the issue with the delay period timing. Why are the figures still showing a time course of 10 seconds for delay when the majority of experiments have the choice phase begin after 9 seconds, though?

Point 2/7:

Same response from the authors for both points. As far as your methods section allows, I understand that the fluorescence data was subjected to the same normalisation to background (as is common). I also now measured the magnitude of the responses in the rightmost panel in Fig1c and see that there is a small difference between error and correct responses. You say you normalised the right panels to the baseline. But the traces in epochs -5 to 0 are still different. What did you normalise?

Also, may I encourage the authors to add labels to some panels, e.g. Fig 1c/d vis-a-vis LCA3 and RCA3 if you do not want to use different colours (for RCA and LCA data).

Point 3:

Well addressed with additional panels in Supp Fig.

Point 4:

Well addressed with additional info in Methods.

Point 5:

Thank you for the response. Please add this / additional detail to the MS to make clear that unit data was pooled across animals, possibly also in the Methods. You currently only mention this in some Figure Legends (c.f. point 12).

Point 6:

Fine.

Point 7:

See above.

Point 8:

Fine.

Point 9:

Fine.

Point 10:

Fine.

Point 11:

Fine.

Point 12:

Still not clear, see above.

Point 13:

Fine.

Point 14:

Still a lot of issues. Scale bars from many histology images missing. Missing axes labels and units (e.g. the colour bars for 3D plots, please state what the scale measures, even if A.U.).

Point 15:

Some of the stats are still very confusing: post-hoc tests are used where not needed; asterisks indicating significance in figures do not match up with figure legends (e.g. Fig. 3b, what are the values corresponding to "*" and "**")? The legend only mentions one test with low significance for both ANOVA and post-hoc test). Please revise.

Point 16:

The methods have greatly improved in places, however, still lack the organisation, style, and detail commonly expected and, critically, required to reproduce experiments. You may want to refer to published guides.

Point 17:

I appreciate the additional figure but my questions remain unanswered: Such different light intensities can have very different effects. how was this measured and calibrated? with reference to the (rich) literature on this issue, what are the potential confounds of prolonged light stimulation. how is the interpretation of data affected by the limitation of the manipulation to specific time points (i.e. your main finding concerns the moment a choice is made but your manipulation arguably starts before and/or ends after)?

Overall, the manuscript improved and, following provision of more detail, I now have greater confidence in the findings. However, the manuscript remains difficult to read and important controls/discussion of potential confounds is missing. A very thorough revision and proofreading would be required before any publication.

Reviewer #2 (Remarks to the Author):

Re-review Comments:

The authors did a very good job addressing this reviewer's comments. I endorse its publication.

Description of the study:

In this study, Song and col. attempt to unveil the contribution of left and right CA3 neural activity and cell properties, afferent circuits and cell types that support spatial working memory (SWM) tasks. Through a massive number of experiments using behavioral, functional, tracing and optogenetic approaches, the authors conclude that GABAergic drive from medial septum to the left hippocampus modulates neural activity associated with a correct prediction to solve a SWM task.

Reviewer #3 (Remarks to the Author):

I thank the authors for addressing my comments and clarifying what was previously unclear. I'm happy for this manuscript to be published after the following:

The authors have addressed my first major comment satisfactorily. As for the second major comment,

the authors should at least discuss the possibility of the effect of silencing MS inputs being permissive rather than instructive for working memory in the manuscript itself.

With the ANOVAs in the text, the authors should report main effects and not just interactions (e.g. effect of light and effect of group in figure 3b)

The explanation for using 20 hz stimulation rate should be made in the manuscript itself (in detail in methods section and briefly in main text)

Dear reviewers,

Thanks again for your valuable comments on our revised manuscript. Firstly, we appreciate a lot for your supports on our work. Secondly, we felt so sorry that a few questions were not answered very satisfactorily. Regarding the reviewer 1 and reviewer 3's further comments, we studied and researched very carefully, and have responded to those questions one by one and make corrections which we hope to satisfy the reviewers and meet with their approval. Please check all the corrections in the revised manuscript and the responses to the reviewers' comments are as follows (the replies are highlighted in red).

The detailed responses to referees' comments are listed below.

Reviewers' comments:

Reviewer #1 (Remarks to the Author):

I thank the authors for their point-by-point response to my earlier comments. I shall reply in kind:

Point 1:

I appreciate the response figure which shows time courses for the entire dataset. This is very informative. I encourage the authors to include this either in Fig 1 or in a supplementary figure. Thank you also for clarifying the issue with the delay period timing. Why are the figures still showing a time course of 10 seconds for delay when the majority of experiments have the choice phase begin after 9 seconds, though?

Response to the reviewer: Thanks very much for this comment. We are sorry that our explanation in the 1st edition of revision caused misunderstanding. Regarding this issue, we would like to explain that: we recorded a total of 14 mice for T-maze tasks. The delay phases were counted for 10s in each trial in the majority of mice; while, only a minority of mice, including the number of 3, 11, 14, a few trials of their delay phases were counted for less than 10s, but still roughly for between 9s and 10s. Due to the fact that the majority of experiments were recorded for 10s delay, that is why the figures showed a time course of 10s rather than 9s.

In addition, as suggested by the reviewer, this data has been placed in to the supplementary figure 1h (please refer to the line 847) in the revised manuscript.

Point 2/7:

Same response from the authors for both points. As far as your methods section allows, I understand that the fluorescence data was subjected to the same normalisation to background (as is common). I also now measured the magnitude of the responses in the rightmost panel in Fig1c and see that there is a small difference between error and correct responses. You say you normalised the right panels to the baseline. But the traces in epochs -5 to 0 are still different. What did you normalise? Also, may I encourage the authors to add labels to some panels, e.g. Fig 1c/d vis-a-vis LCA3 and RCA3 if you do not want to use different colours (for RCA and LCA data).

Response to the reviewer: Thank you very much for this valuable comment. We felt so sorry that our imprecise description caused your confusion about this. "Normalization" is not a precise word to describe the calculation, what we should do is calculating the variation between the fluorescent data with the baseline. The calculating formula was used as listed as following:

Signal Variation = MeanSignal (1, 2) - MeanSignal(-5, -1).

Mean signal (1, 2) stands for the mean increased signal in the time course from 1s to 2s, and Mean signal (-5, -1) stands for the mean signal of baseline in the time course from -5s to -1s. We hope this could help better understand the data we present.

In addition, Figure1c/d was relabeled and revised as suggested.

Point 3:

Well addressed with additional panels in Supp Fig.

Response to the reviewer: Thank you very much for this comment. The related contents has been added in the manuscript (please refer to the line 153-155) and the figure in the supplementary figure 1g (please refer to the line 847) as suggested in the revised manuscript.

Point 4:

Well addressed with additional info in Methods.

Response to the reviewer: Thank you very much for this comment. The detailed illustrations and the generation of z-scores have been added in methods (please refer to the line 591-596) in the revised manuscript.

Point 5:

Thank you for the response. Please add this / additional detail to the MS to make clear that unit data was pooled across animals, possibly also in the Methods. You currently only mention this in some Figure Legends (c.f. point 12).

Response to the reviewer: Thank you very much for this comment. As suggested, we have added details of the numbers of animals used in methods; please refer to the line 579-580 in the revised manuscript.

Point 6:

Fine.

Point 7:

See above.

Point 8:

Fine.

Point 9:

Fine.

Point 10:

Fine.

Point 11:

Fine.

Point 12:

Still not clear, see above.

Response to the reviewer: Thank you very much for this. The details about animal information have been added in methods; please refer to the line 579-580 in the revised manuscript.

Point 13:

Fine.

Point 14:

Still a lot of issues. Scale bars from many histology images missing. Missing axes labels and units (e.g. the colour bars for 3D plots, please state what the scale measures, even if A.U.).

Response to the reviewer: Thanks the reviewer for pointing this out. We regret for our carelessness on missing the scale/axes labels. We have revised these figures and added scale bars in figure legends in the revised manuscript. Please refer to the line of 895, 906, 921, 1023-1024, 1041,1044, and 1085,1094 in the revised manuscript.

Point 15:

Some of the stats are still very confusing: post-hoc tests are used where not needed; asterisks indicating significance in figures do not match up with figure legends (e.g. Fig. 3b, what are the values corresponding to "*" and "***"? The legend only mentions one test with low significance for both ANOVA and post-hoc test). Please revise.

Response to the reviewer: Thanks the reviewer for pointing this out. We have revised as the reviewer suggested. Please refer to the figure legends in the revised manuscript.

Point 16:

The methods have greatly improved in places, however, still lack the organisation, style, and detail commonly expected and, critically, required to reproduce experiments. You may want to refer to published guides.

Response to the reviewer: Thanks very much for this comment. We are sorry for missing some details. Following the published guides, we have added related information (the origin of products we used in the experiments) in the

revised manuscript, which would benefit for other researchers if they would like to reproduce the results.

Point 17:

I appreciate the additional figure but my questions remain unanswered: Such different light intensities can have very different effects. how was this measured and calibrated? with reference to the (rich) literature on this issue, what are the potential confounds of prolonged light stimulation. how is the interpretation of data affected by the limitation of the manipulation to specific time points (i.e. your main finding concerns the moment a choice is made but your manipulation arguably starts before and/or ends after)?

Response to the reviewer: thanks very much for this valuable comment. We are so sorry that our answers were not satisfying. Regarding the reviewer's new questions, we would like to provide more details about the utility of opto-fibers in our optogenetic experiments. The transmissivity of every optic fiber was tested before use, which was approximately 80%, and the opto-intensity of each opto-fiber was measured and maintained at a certain degree (590nm of laser at a power of 8-10 mW; 470nm of laser at a power of 1-2 mW). These parameters were selected for use based on many high quality published literatures (Felix-Ortiz, Beyeler et al. 2013, Felix-Ortiz, Burgos-Robles et al. 2016, Allsop, Wichmann et al. 2018).

Certain studies have suggested that higher intensive laser light or prolonged stimulation by laser light can produce heat in brain tissue and thereby lead to inhibition/damages of the neuronal activity as well as behavioral ability in the absence of opsins (Cardozo Pinto and Lammel 2019, Owen, Liu et al. 2019). Regarding this, we limited the time durations of laser stimulation between 3 s and 5 s. What's of importance, the time durations of laser stimulation could cover the initiation of the choice phase (which means it can cover the time points of firing of those prediction and main arm neurons), which can avoid the side effects caused by the high intensity and long-lasting stimulation of laser light.

In our optogenetic experiments, during the sample and the choice run, laser light was turned on at the same time as the door was open, so 590nm/470nm of laser manipulation could cover the beginning of the initiation of the choice run. In particular, 590nm of laser was lasted for 5 second for the activation of

CA3 neurons, while 470nm of laser was lasted for 3 second (rather than 5 second) for the inhibition of CA3 neurons, since 5 second of inhibition would cause the epilepsy seizure of mice.

We thank you again for your valuable comments that help us illustrate our work more precise and have greatly improved the quality of our manuscript. We appreciate a lot for your support on our work.

References:

Felix-Ortiz, A. C., A. Beyeler, C. Seo, C. A. Leppla, C. P. Wildes and K. M. Tye (2013). "BLA to vHPC inputs modulate anxiety-related behaviors." Neuron **79**(4): 658-664.

Felix-Ortiz, A. C., A. Burgos-Robles, N. D. Bhagat, C. A. Leppla and K. M. Tye (2016). "Bidirectional modulation of anxiety-related and social behaviors by amygdala projections to the medial prefrontal cortex." Neuroscience **321**: 197-209.

Allsop, S. A., R. Wichmann, F. Mills, A. Burgos-Robles, C. J. Chang, A. C. Felix-Ortiz, A. Vienne, A. Beyeler, E. M. Izadmehr, G. Guber, M. I. Cum, J. Stergiadou, K. K. Anandalingam, K. Farris, P. Namburi, C. A. Leppla, J. C. Weddington, E. H. Nieh, A. C. Smith, D. Ba, E. N. Brown and K. M. Tye (2018). "Corticoamygdala Transfer of Socially Derived Information Gates Observational Learning." Cell **173**(6): 1329-1342 e1318.

Cardozo Pinto, D. F. and S. Lammel (2019). "Hot topic in optogenetics: new implications of in vivo tissue heating." Nat Neurosci **22**(7): 1039-1041.

Etter, G., S. van der Veldt, F. Manseau, I. Zarrinkoub, E. Trillaud-Doppia and S. Williams (2019). "Optogenetic gamma stimulation rescues memory impairments in an Alzheimer's disease mouse model." Nat Commun **10**(1): 5322.

Overall, the manuscript improved and, following provision of more detail, I now have greater confidence in the findings. However, the manuscript remains difficult to read and important controls/discussion of potential confounds is missing. A very thorough revision and proofreading would be required before any publication.

Reviewer #2 (Remarks to the Author):

Re-review Comments:

The authors did a very good job addressing this reviewer's comments. I

endorse its publication.

Description of the study:

In this study, Song and col. attempt to unveil the contribution of left and right CA3 neural activity and cell properties, afferent circuits and cell types that support spatial working memory (SWM) tasks. Through a massive number of experiments using behavioral, functional, tracing and optogenetic approaches, the authors conclude that GABAergic drive from medial septum to the left hippocampus modulates neural activity associated with a correct prediction to solve a SWM task.

Response to the reviewer: We thank you again for your valuable comments that have greatly improved the quality of our manuscript. We appreciate a lot for your support on our work.

Reviewer #3 (Remarks to the Author):

I thank the authors for addressing my comments and clarifying what was previously unclear. Im happy for this manuscript to be published after the following:

point2

The authors have addressed my first major comment satisfactorily. As for the second major comment, the authors should at least discuss the possibility of the effect of silencing MS inputs being permissive rather than instructive for working memory in the manuscript itself.

Response to the reviewer: Thanks very much for the reviewer pointing this out. After furthering researching published papers about MS and hippocampus, we made discussions about the possibility of the effects of silencing MS inputs on working memory as following:

“The MS/DB has been emerged as a key modulator for hippocampal function via septohippocampal pathway. Both GABAergic and cholinergic neurons in MS/DB are correlated with learning and memory and hippocampal rhythmogenesis (Berke, Hetrick et al. 2008, Roland, Stewart et al. 2014). Earlier studies have demonstrated that the septal cholinergic neurons project diffusely to both the interneurons and pyramidal neurons in hippocampus,

which is contrast with the GABAergic MS/DB neurons that exclusively innervate hippocampal interneurons (Frotscher and Leranth 1985, Hangya, Borhegyi et al. 2009). Beck H team revealed that cholinergic MS/DB neurons could impact on hippocampal circuits through two distinct pathways. One is a direct pathway that septo-hippocampal cholinergic projection leads to increased firing of hippocampal inhibitory interneurons while decreased firing of pyramidal cells, and the other one is indirect that PV neurons in MS/DB would be recruited by cholinergic neurons to selectively innervate hippocampal interneurons and be responsible for precise synchronization of hippocampal networks (Roland, Stewart et al. 2014). In addition, a somatic MS GABAergic inputs were discovered to be able to target to hippocampal axo-axonix cells to modulate hippocampal sharp waves and project to CA3 (Viney, Lasztocki et al. 2013). Dannerberg H team found that GABAergic lesions of the MS/DB in rats could impair the hippocampal acetylcholine efflux and the spatial working memory in the DNMTF tasks (Dannenberg, Pabst et al. 2015). Recent study also demonstrated that optogenetic stimulation of MS-PV neurons could recover slow gamma oscillations in hippocampus and rescue the spatial memory of AD model mice (Etter, van der Veldt et al. 2019).

In our data, we mapped the anatomical specificity and organization of MS to unilateral CA3 with distinct neuronal types. We revealed that MS cholinergic neurons mainly project to bilateral CA3 pyramidal neurons, while PV neurons in MS project to unilateral CA3 neurons. Selective suppression of MS^{PV}-LCA3 projecting neurons impaired the performance of SWM during the choice phase, suggesting that PV neurons in MS have potential to dominate the lateralization of LCA3 in SWM.

Regarding the global effects of MS GABAergic inputs to Hippocampus, we speculated that MS GABAergic inputs would similarly regulate hippocampal CA3 via manipulating its oscillatory patterns, which would consequently influence the process of memory, and this influence might be permissive. Further exploration would focus on revealing whether the lateralization of LCA3 would be correlated with the oscillatory modulation of MS GABAergic inputs to hippocampal CA3 and what types of CA3 neurons would be involved in this modulation”.

Please refer to the line414-447 in the revised manuscript.

References:

Berke, J. D., V. Hetrick, J. Breck and R. W. Greene (2008). "Transient 23-30 Hz oscillations in mouse hippocampus during exploration of novel environments." Hippocampus **18**(5): 519-529.

Roland, J. J., A. L. Stewart, K. L. Janke, M. R. Gielow, J. A. Kostek, L. M. Savage, R. J. Servatius and K. C. Pang (2014). "Medial septum-diagonal band of Broca (MSDB) GABAergic regulation of hippocampal acetylcholine efflux is dependent on cognitive demands." J Neurosci **34**(2): 506-514.

Frotscher, M. and C. Leranth (1985). "Cholinergic innervation of the rat hippocampus as revealed by choline acetyltransferase immunocytochemistry: a combined light and electron microscopic study." J Comp Neurol **239**(2): 237-246.

Hangya, B., Z. Borhegyi, N. Szilagyi, T. F. Freund and V. Varga (2009). "GABAergic neurons of the medial septum lead the hippocampal network during theta activity." J Neurosci **29**(25): 8094-8102.

Dannenberg, H., M. Pabst, O. Braganza, S. Schoch, J. Niediek, M. Bayraktar, F. Mormann and H. Beck (2015). "Synergy of direct and indirect cholinergic septo-hippocampal pathways coordinates firing in hippocampal networks." J Neurosci **35**(22): 8394-8410.

Viney, T. J., B. Lasztocki, L. Katona, M. G. Crump, J. J. Tukker, T. Klausberger and P. Somogyi (2013). "Network state-dependent inhibition of identified hippocampal CA3 axo-axonic cells in vivo." Nat Neurosci **16**(12): 1802-1811.

Etter, G., S. van der Veldt, F. Manseau, I. Zarrinkoub, E. Trillaud-Doppia and S. Williams (2019). "Optogenetic gamma stimulation rescues memory impairments in an Alzheimer's disease mouse model." Nat Commun **10**(1): 5322.

With the ANOVAs in the text, the authors should report main effects and not just interactions (e.g. effect of light and effect of group in figure 3b)

Response to the reviewer: Thanks very much for this comment. We have done the revision as suggested. Please refer to the figure legends (figure 3 and supplementary figure 3) in the revised manuscript.

The explanation for using 20 hz stimulation rate should be made in the manuscript itself (in detail in methods section and briefly in main text)

Response to the reviewer: Thanks very much for this suggestion. We have placed the details of 20Hz stimulation in methods (please refer to the line 680-682) and the explanation of using this frequency in the discussion (please refer to the line 457-465) in the revised manuscript.

Again, thanks very much for these precious comments that have greatly improved the quality of our manuscript. We appreciate a lot for your support on our work.

Dear reviewer,

Thanks again for your valuable comments on our revised manuscript. Firstly, we appreciate a lot for your support on our work. Secondly, we really apologize that we might not fully get the points of the reviewer and a few questions were not answered very satisfactorily. Regarding the reviewer 1's further comments, together with journal editor's requirements, we revised our whole manuscript and have responded to the questions one by one which we hope to satisfy the reviewer and meet with the approval for publication.

The detailed responses to referees' comments are listed below.

REVIEWERS' COMMENTS:

Reviewer #1 (Remarks to the Author):

Point 1: Addressed, albeit the MS still lacks clarity regarding this. Timing of responses is critical for the interpretation of the data and most readers will not have the opportunity to quiz the authors. It is good to see the 3D plots included in SuppFig1, however, see point 14 below.

Response: Thanks so much for this. The performance of T-maze task that was done manually was mentioned in the methods as suggested.

Point 2/7: Thank you for the explanation of how you normalised the traces in Fig1c/d and giving some idea of how the "quantification" in the bar charts (which *still* lack units) was performed. I could not find this info anywhere in the manuscript despite feeling other

readers would also like to know how you quantified key effects of your study. Would you consider including a clear description in the methods?

Response: thanks very much for this suggestion. Following the advice by reviewer, we have added the following information for quantification of signal variation in the methods:

“Signal variation of GCamP6s in choice phases was calculated as follows:

Signal Variation = Mean Signal (1, 2) - Mean Signal (-5, -1).

Mean signal (1, 2) stands for the mean increased signal in the time course from 1s to 2s, and Mean signal (-5, -1) stands for the mean signal of baseline in the time course from -5s to -1s.”

And we add appropriate units on the figure 1c/d.

Please refer to line of 601-605 in the 3rd revised manuscript.

The reorganisation of the figure now resulted in the panels being described in the Results out of order (a, b, f, e, d, g, c, d). That is rather confusing. Can this be fixed?

Response: thanks for this. We have reorganized the figure 1 as suggested.

Point 12: Addressed.

Response: thanks for this.

Point 14: My last comment was: “Still a lot of issues. Scale bars from many histology images missing. Missing axes labels and units (e.g. the colour bars for 3D plots, please

state what the scale measures, even if A.U.).” The authors *only* addressed the issues of missing / unclear scale bars. There are still a lot of axes without labels and units. I am slightly despairing over this at this stage, given the authors had multiple opportunities to rectify this. I repeat what I said before, the present study offers a plethora of data. However, these data remain inaccessible to proper scrutiny and study by the reader if methods are not (sufficiently and clearly) described and measurements are left ambiguous for lack of labels. I can still count more than a dozen plots lacking appropriate annotation. Wherever there is data plotted, it requires a label/description/scale bars. Any data without is meaningless to the reader.

Response: thanks very much for the professional attitude and carefulness of reviewer. We really apologize that we might not fully understand the reviewer's points before. Following this comment, we have revised every figure exactly following the journal's data policy and requirements and ensured that labels, units, scale bars and other related information have been marked and added in the figures or figure legends. Please refer to the updated figures and figure legends through the 3rd revised manuscript.

Point 15: I thank the authors for including more detail on the stats. The labelling is still somewhat idiosyncratic (e.g. “*” is used for two different comparisons in the same plot, such as 3b). But it is now clear the stats are present. For final recommendations on how to present statistical comparisons in the figures and legends please consult with the journal editors / style guide.

Response: thanks so much for this. We have revised our data following the journal editors.

Point 16: Addressed.

Response: thanks so much for this.

Point 17: I thank the authors for their reply. I regret they chose not to include some discussion of the potential confounds in their manuscript. I also hoped to find more clarity in the Methods (e.g. as I previously suggested and is common in the literature, something to the effect of, “we measured light intensity at the laser launch, tip of the coupled fibre (type, supplier of fibre), and the fibre ferrule implant using a x-type power meter. Intensity at the implanted fibre tip was thus typically xx% lower than at the coupled fibre end. We measured intensity at the coupled fibre before connecting to the implanted fibre ferrule before connecting each animal every day and kept intensity adjusted to x-xx mW”).

Response: thanks so much for this valuable comment. As suggested, we discussed the potential confounds of the intensive light and prolonged stimulation, which has been added in the discussion part in the revised manuscript. Please refer to line of 406-424 in the 3rd revised manuscript.

Also, we added detailed description of measurement of optic fibre transmissivity in the methods as following :” Measurement of optic fibre transmissivity: we measured light intensity at the laser launch, tip of the coupled fibre (diameter, 200 µm; Ferrule O.D, 1.25 mm; N.A., 0.37; length, 3.0 or 4.0 mm; Inper Inc., China), and the fibre ferrule implant by

optical power meter (PM121D, thorlab, USA). Fibre ferrules were sorted based on optical transmissivity (>80%). We measured intensity at the coupled fibre before connecting to the implanted fibre ferrule and connecting each animal every day. The intensity was kept to adjust to certain power (ChR2/ChETA stimulation: 1mW; NpHR3.0 stimulation:10mW; fibre photometry: 40 μ W)". Please refer to the line of 522-529 in the 3rd revised manuscript.

Again, we sincerely apologize for our carelessness or misunderstandings during last revision. We hope this revision would be satisfactory with the reviewer. Thanks again for your support and your valuable comments that help improve the quality of our manuscript dramatically.